# Comprehensive T cell repertoire characterization of non-small cell lung cancer

Alexandre Reuben ⓘD et al.#

Immunotherapy targeting T cells is increasingly utilized to treat solid tumors including non-small cell lung cancer (NSCLC). This requires a better understanding of the T cells in the lungs of patients with NSCLC. Here, we report T cell repertoire analysis in a cohort of 236 early-stage NSCLC patients. T cell repertoire attributes are associated with clinicopathologic features, mutational and immune landscape. A considerable proportion of the most prevalent T cells in tumors are also prevalent in the uninvolved tumor-adjacent lungs and appear specific to shared background mutations or viral infections. Patients with higher T cell repertoire homology between the tumor and uninvolved tumor-adjacent lung, suggesting a less tumor-focused T cell response, exhibit inferior survival. These findings indicate that a concise understanding of antigens and T cells in NSCLC is needed to improve therapeutic efficacy and reduce toxicity with immunotherapy, particularly adoptive T cell therapy.

---

# A full list of authors and their affiliations appears at the end of the paper.

N SCLC bears a high mutational load[1,2] which has been linked to tumor-specific antigens, termed neoantigens, that may activate host anti-tumor T cell responses[3,4]. This has led to renewed excitement for therapies targeting the T cell repertoire, such as checkpoint blockade using cytotoxic T lymphocyte-associated antigen-4 (CTLA-4)[5], programmed death-1 (PD-1)[6], and programmed death ligand-1 (PD-L1)[7], as well as personalized mutation-specific vaccines[8] and T cell-based therapies[9]. Therefore, there is a considerable need to better understand the tumor-infiltrating T lymphocyte (TIL) repertoire. Targeted sequencing of the highly variable CDR3 region of the beta chain of the T cell receptor (TCR) can be used to identify T cell clones, their frequencies, and the existence of antigenic responses within a repertoire[10]. It has been suggested that patients with greater T cell clonal expansion (clonality), a characteristic of antigenic responses, have improved clinical responses to immune checkpoint inhibitors in melanoma[11,12]. However, T cell responses may vary in their kinetics and distribution in different organs, which complicates T cell repertoire analysis. This is further compounded in lung cancer due to the local inflammation linked to smoking exposure and non-tumor-associated pathogens[13]. Recent studies have investigated protective immune responses in the lung by defining the role of neutrophils, antigen-presenting cells (APCs), and T cells[14,15]. However, little is known about the attributes of the T cell repertoire and how they relate to patient outcome. Here, we delineate the T cell repertoire in a cohort of 236 localized NSCLC patients, 11 chronic obstructive pulmonary disease (COPD) patients without lung cancer, and 24 healthy lung donors (Table 1) to define the relationship between the T cell repertoire and tumor clinicopathologic features as well as the tumor immunogenomic landscape and to determine its impact on patient survival in early stage NSCLC. We show that a significant proportion of T cells are shared between the tumor and adjacent lung, and that these T cells may be enriched for their ability to recognize shared mutations throughout the lung or viruses. We also demonstrate that patients with a less tumor-focused T cell repertoire exhibit worse outcome.

## Results

**T cell clonality is associated with CD8 T cells**. To study the attributes of the T cell repertoire in localized lung cancers, we performed next generation sequencing of the CDR3 variable region of the beta chain of the TCR involved in antigen binding

from 236 resected tumors from treatment-naïve NSCLC patients[16,17]. T cell density, an estimate of T cell infiltration in the tumor, ranged from 0.01 to 1.0 ($n = 225$; average = 0.24) (Supplementary Fig. 1A), while richness, a measure of T cell diversity ranged from 204 to 20,479 unique T cell rearrangements ($n = 224$; average = 5335 unique rearrangements) in this cohort of NSCLC tumors (Supplementary Fig. 1B). T cell clonality ranged from 0.06 to 0.36 ($n = 225$; average = 0.15) (Supplementary Fig. 1C). Overall, T cell density was positively correlated with richness and clonality (Density vs Richness: $r = 0.75$, $p < 0.0001$; Density vs Clonality: $r = 0.20$, $p = 0.003$, Spearman rank correlation, Supplementary Fig. 1D–E). However, richness and clonality were inversely correlated, highlighting that overall, a more diverse T cell infiltrate may be suggestive of lower reactivity ($r = -0.16$, $p = 0.019$, Spearman rank correlation, Supplementary Fig. 1F).

To define the phenotype of T cells comprising the T cell repertoire in these tumors, we reanalyzed our recently published T cell profiling data from immunohistochemical staining (IHC)[18] as well as gene expression data[19,20] from the same cohort. Analysis of IHC data for 8 T cell-associated markers: CD3 (T cells), CD4 (helper T cells), CD8 (cytotoxic T cells), FoxP3 (regulatory T cells), CD45RO (antigen-experienced T cells), Granzyme B (cytotoxic T cells), and PD-1 (activated/dysfunctional T cells) as well as PD-L1 (Supplementary Fig. 2A) demonstrated that T cells in this cohort of NSCLC tumors were predominantly CD4-positive with an average CD4:CD8 ratio of 1.65 (ranging from 0.3 to 5.3) ($n = 146$; Supplementary Fig. 2B–C), consistent with prior work from our group[21]. Though all T cell markers were positively correlated, the density of CD4 T cells was most highly correlated with the density of FoxP3 ($n = 146$; $r = 0.63$, $p < 0.0001$, Spearman rank correlation, Supplementary Fig. 2D). Alternately, CD8 T cell density correlated most strongly with GzmB ($n = 146$; $r = 0.76$, $p < 0.0001$, Spearman rank correlation, Supplementary Fig. 2E), highlighting the cytotoxic potential of this subset.

When evaluating the relationship between immune markers and the tumor T cell repertoire, T cell density and richness were correlated with CD3 ($r = 0.53$; $p < 0.0001$ and $r = 0.30$; $p = 0.0004$, Spearman rank correlation), CD4 ($r = 0.39$; $p < 0.0001$ and $r = 0.33$; $p < 0.0001$, Spearman rank correlation), and CD8 ($r = 0.51$; $p < 0.0001$ and $r = 0.27$, $p = 0.002$, Spearman rank correlation), as anticipated ($n = 146$; Supplementary Fig. 3A–F). However, T cell clonality correlated only with CD3 ($n = 135$; $r = 0.24$; $p = 0.005$, Spearman rank correlation, Fig. 1a), and CD8 ($r = 0.30$; $p = 0.0003$ Spearman rank correlation), but not with CD4 ($n = 135$; $r = -0.03$; $p = 0.753$, Spearman rank correlation, Fig. 1b–c), highlighting the greater proliferative potential of CD8 T cells and suggesting T cell clonality may be mainly driven by the clonal expansion of CD8-positive T cells[22]. Importantly, analysis of RNA expression from these tumors[20] demonstrated that T cell clonality was also positively correlated with GzmB ($n = 141$; $r = 0.47$; $p < 0.0001$, Spearman rank correlation, Fig. 1d) and IFN-γ expression ($n = 141$; $r = 0.52$; $p < 0.0001$, Spearman rank correlation, Fig. 1e), further capturing the activated and cytotoxic phenotype of CD8 T cells following antigen encounter.

We further investigated the relationship between T cell phenotype and repertoire by comparing T cell density, richness, and clonality in tumors with high versus low CD45RO (antigen-experienced T cells) and PD-1 (activated T cells). As shown in Fig. 2, CD45RO$^{hi}$ tumors exhibited higher T cell infiltration ($n = 135$; $p = 0.0016$, Mann–Whitney test), and moderately increased richness ($n = 134$; $p = 0.0851$, Mann–Whitney test) but no difference in clonality ($n = 135$; $p = 0.5027$, Mann–Whitney test) when compared to their CD45RO$^{lo}$ counterparts (Fig. 2a–c). However, analysis of PD-1$^{hi}$ tumors demonstrated higher T cell

### Table 1 Clinicopathologic features of studied subjects.

| Category | NSCLC ($n = 236$) | COPD ($n = 11$) | Healthy ($n = 24$) |
|---|---|---|---|
| Age (yr) | 66.3 ± 9.9 | 61.7 ± 8.8 | 38 ± 16.8 |
| *Gender* | | | |
| Female | 107 (45) | 2 (18) | 12 (50) |
| Male | 129 (55) | 9 (82) | 12 (50) |
| *Tumor type* | | | |
| ADCA | 146 (62) | NA | NA |
| SCCA | 89 (37) | NA | NA |
| ADCA/SCCA | 1 (1) | NA | NA |
| *Stage* | | | |
| Stage I | 114 (48) | NA | NA |
| Stage II | 79 (33) | NA | NA |
| Stage III | 43 (19) | NA | NA |
| *Smoking status* | | | |
| Current | 102 (43) | 1 (9) | 11 (46)$^a$ |
| Former | 114 (48) | 10 (91) | NA |
| Never | 20 (9) | 0 | 8 (33) |
| Unknown | 0 | 0 | 5 (21) |

$^a$Healthy lung samples are classified by smoker or non-smoker only.

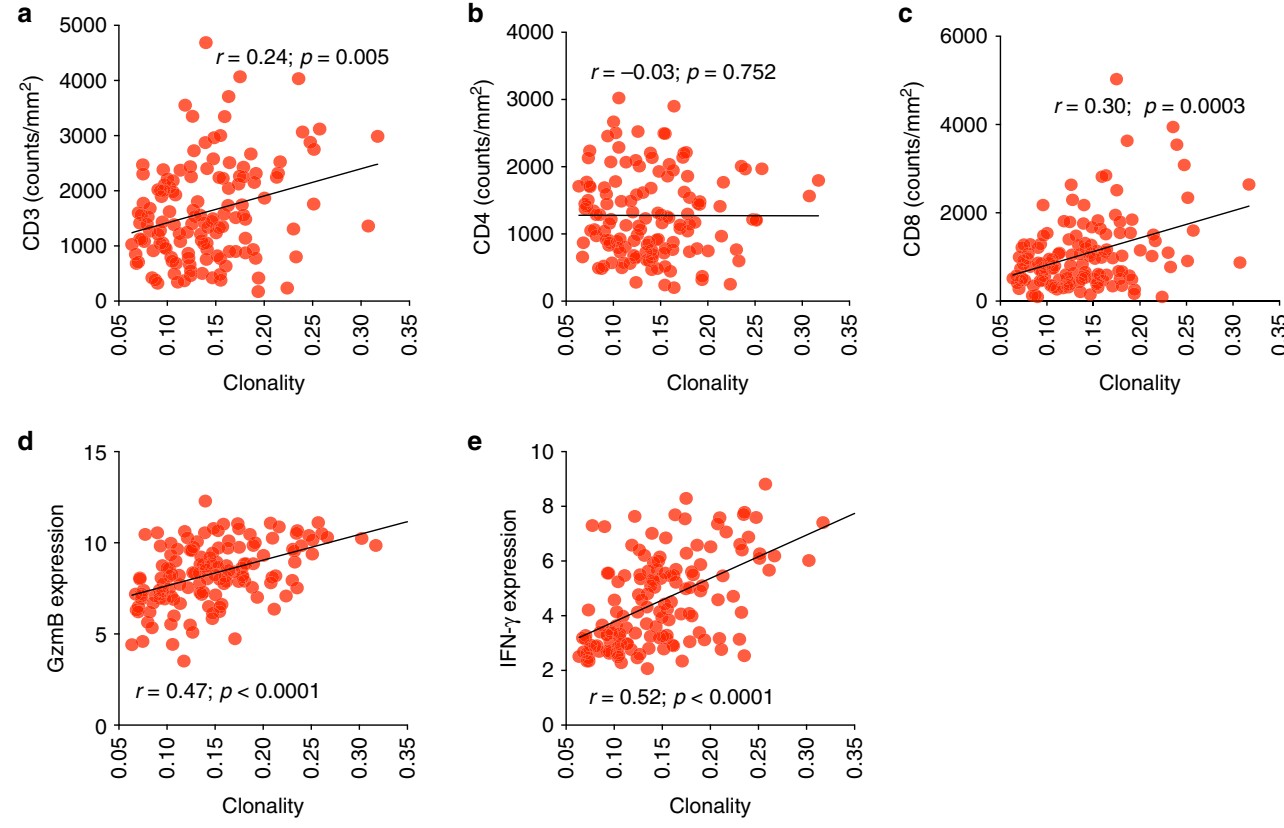

**Fig. 1 TCR clonality is associated with CD8 T cell function.** Correlation between T cell clonality and (**a**) CD3 density ($n = 135$), (**b**) CD4 density ($n = 135$) and (**c**) CD8 density ($n = 135$) by IHC as well as (**d**) GzmB expression ($n = 141$), (**e**) IFN-γ expression ($n = 141$) by gene expression profiling.

density ($n = 135$; $p < 0.0001$, Mann–Whitney test), richness ($n = 134$; $p = 0.0083$, Mann–Whitney test) and clonality ($n = 135$; $p = 0.0104$, Mann–Whitney test) than their PD-1$^{lo}$ counterparts (Fig. 2d–f).

The distribution of T cells within the tumor microenvironment can be suggestive of an efficacious anti-tumor T cell response, with several groups showing enrichment for T cells at the periphery or center is related to improved outcome[11,23]. As such, we evaluated the relation between enrichment of T cell markers CD3, CD4, and CD8 at the tumor center versus periphery, and how this relates to attributes of the T cell repertoire. Overall, no trends were observed with CD3 and CD8 spatial distribution, though higher CD4 at the periphery of the tumor was associated with lower T cell richness ($p = 0.0006$, Mann–Whitney test) and higher T cell clonality ($p = 0.0120$, Mann–Whitney test) at the tumor center by CDR3 sequencing ($n = 119$; Supplementary Fig. 4A–I). This suggests accumulation of CD4 T cells, particularly regulatory T cells, in the tumor center could impair the ability of CD8 T cells to expand in response to antigens (i.e., drive up clonality). Multi-region sequencing was not performed in this cohort, therefore we cannot exclude the possibility that intratumor heterogeneity may have played a role, as shown previously by our group and others[21,24,25].

**Tumor mutational burden is correlated with T cell activation.** The tumor mutational burden (TMB) contributes to immunogenicity through the generation of neoantigens targeted by T cell responses. Accordingly, a higher TMB has been reported to be associated with a higher response rate and favorable survival in lung cancer patients across multiple clinical trials[26,27]. Therefore, we first reanalyzed our recently published whole exome sequencing data in the same patient cohort[17] to evaluate the relationship

between the TMB and attributes of the T cell repertoire. On average, 176 (ranging from 3 to 857) nonsynonymous exonic mutations (NSEM) per tumor were identified ($n = 215$; Supplementary Fig. 5A). Higher TMB was correlated with higher T cell clonality ($r = 0.19$; $p = 0.015$, Spearman rank correlation), a lower CD4:CD8 ratio ($r = -0.38$; $p = 0.0002$, Spearman rank correlation), and higher GzmB ($r = 0.32$; $p = 0.0019$ by IHC, $r = 0.26$; $p = 0.02$ by gene expression profiling, Spearman rank correlation) ($n = 215$; Fig. 3a and $n = 146$; Supplementary Fig. 5B–G), supportive of the critical role of somatic mutations in enhancing tumor immunogenicity and triggering T cell responses through the generation of neoantigens. However, no link was seen between HLA loss of heterozygosity (HLALOH) and T cell density, richness and clonality suggesting this resistance mechanism may not have played a key role in this group of patients ($n = 164$; Supplementary Fig. 6A–C).

***EGFR* mutation is associated with low T cell clonality.** The discovery of oncogenic driver mutations, which confer growth advantage to cancer cells has improved our understanding of multiple cancers[28]. Recent studies have suggested that these mutations may impact anti-tumor immune responses, which in turn can alter the dynamics of tumor evolution, particularly under immunotherapy[29,30]. Therefore, we next sought to assess the correlation between the presence of canonical oncogenic driver mutations and attributes of the T cell repertoire. These analyses demonstrated that *EGFR*-mutant tumors had significantly higher richness ($p = 0.017$, Mann–Whitney test) and lower T cell clonality ($p = 0.001$, Mann–Whitney test) compared to *EGFR*-wildtype tumors, though no difference was noted in the density of the T cell repertoire ($p = 0.103$, Mann–Whitney test) ($n = 186$; Fig. 3b–e). This is in line with the lower response rate

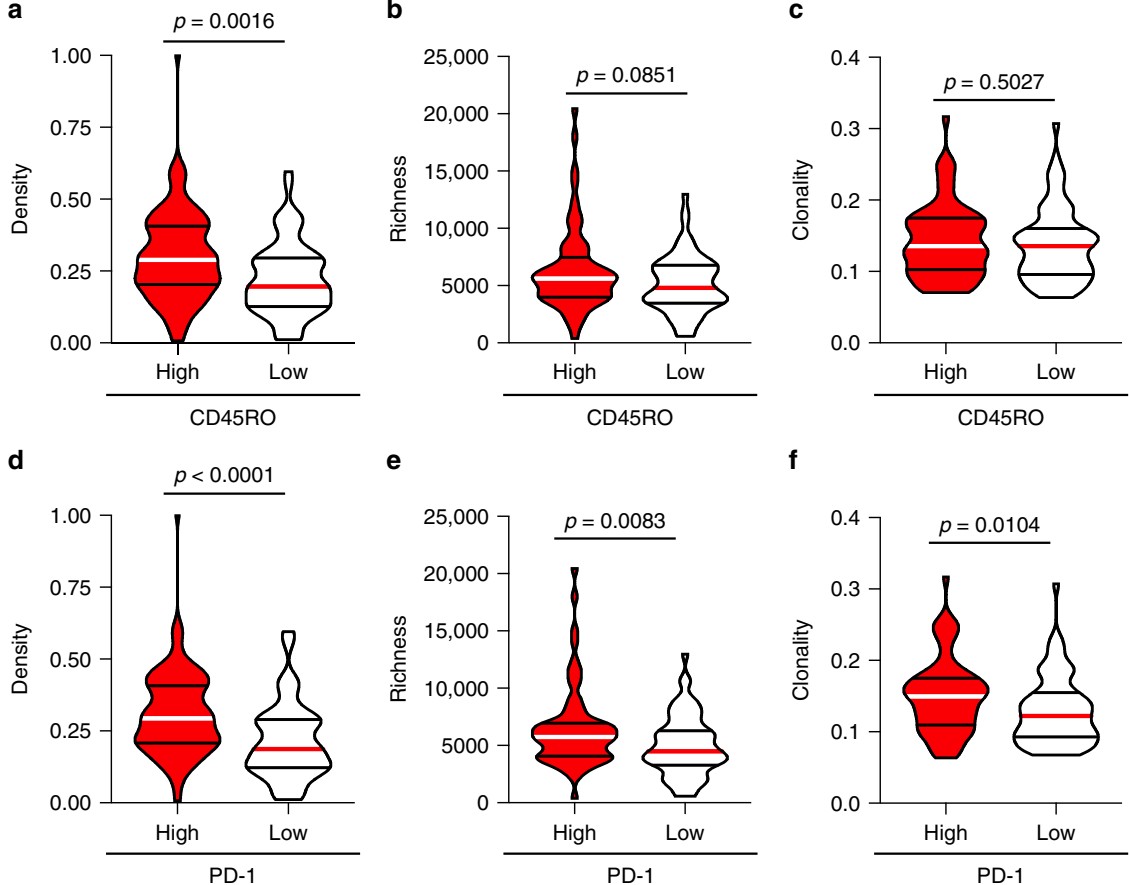

**Fig. 2 T cell density, richness and clonality are enriched in PD-1hi tumors.** (**a**) T cell density (*n* = 135), (**b**) richness (*n* = 134) and (**c**) clonality (*n* = 135) in CD45ROhi and CD45ROlo tumors. (**d**) T cell density (*n* = 135), (**e**) richness (*n* = 134) and (**f**) clonality (*n* = 135) in PD-1hi and PD-1lo tumors. Bars represent median and quartiles.

and survival benefit of *EGFR*-mutant NSCLCs treated with immune checkpoint blockade[31–33] despite the high expression of PD-L1 in many of these tumors[34,35]. The exact molecular mechanisms underlying resistance of *EGFR*-mutant NSCLC to immunotherapy are not yet understood, with a concomitant low mutation load thought to be a major culprit[31–33]. Accordingly, all *EGFR*-mutant patients were found to exhibit a TMB within the bottom tertile of the cohort (*p* < 0.0001, Mann–Whitney test, Fig. 3b). In order to adjust for this difference in TMB, we focused exclusively on $EGFR_{WT}$ tumors within the lowest TMB tertile. Although TMB was comparable between $EGFR_{MUT}$ and $EGFR_{WT}$ TMBlo tumors in this subgroup (*n* = 51; Fig. 3f), T cell clonality remained higher in $EGFR_{WT}$ tumors (*n* = 43; *p* = 0.015, Mann–Whitney test, Fig. 3g–i). These results suggest that $EGFR_{WT}$ tumors could potentially induce better T cell expansion regardless of a low TMB or alternatively that the low TMB in these tumors may have resulted from depletion of immunogenic tumor clones (more likely with higher TMB) by reactive T cells, and as a result driven down the TMB. Conversely, clonality was consistently lower in $EGFR_{MUT}$ tumors, thereby suggesting that T cells may not be expanding (leading to low clonality), most likely due to the existence of alternative immunosuppressive mechanisms, which prevent antigen recognition and T cell expansion. Of note, even within the highest TMB $EGFR_{MUT}$ tumors, no differences were observed in T cell repertoire attributes (*n* = 12; Supplementary Fig. 7A–F). Taken together, these results suggest that there exist TMB-independent mechanisms contributing to the low clonality in $EGFR_{MUT}$ NSCLC tumors. Otherwise, no associations were observed between the T cell repertoire and other frequently mutated cancer genes in NSCLC such as *KRAS* and *TP53*.

**T cell repertoire link to clinicopathologic attributes**. We next sought to assess whether T cell attributes correlated with the clinicopathologic features of these tumors. T cell density and richness showed no differences based on tumor differentiation, though poorly differentiated tumors did exhibit higher T cell clonality than well and moderately differentiated tumors (*n* = 223; *p* = 0.0019 and *p* = 0.0318, respectively; Dunn's multiple comparisons test, Supplementary Fig. 8A–C). T cell density (*n* = 225; *p* = 0.01, Mann–Whitney test) and richness (*n* = 224; *p* = 0.009, Mann–Whitney test) were higher in adenocarcinoma (ADCA) than squamous cell carcinoma (SCCA) (Fig. 4a–b), though clonality was higher in SCCA, in line with prior reports (*n* = 225; *p* = 0.055, Mann–Whitney test, Supplementary Fig. 9A)[15]. This highlights the distinct T cell response to these major histological subtypes of NSCLC. Negative associations between the T cell repertoire and tumor size were also observed, with smaller tumors more densely (*r* = −0.26; *p* = 0.0001, Spearman rank correlation) and diversely (*r* = −0.25; *p* = 0.0002, Spearman rank correlation) infiltrated than their larger counterparts (*n* = 224; Fig. 4c–d). Furthermore, T cell clonality was higher in current and former smokers than in never smokers (*n* = 224; Fig. 4e) and the difference remained statistically significant upon adjustment for TMB using a linear fit model incorporating TMB, demonstrating the impact of cigarette smoke on T cell responses independent of TMB. However, no difference

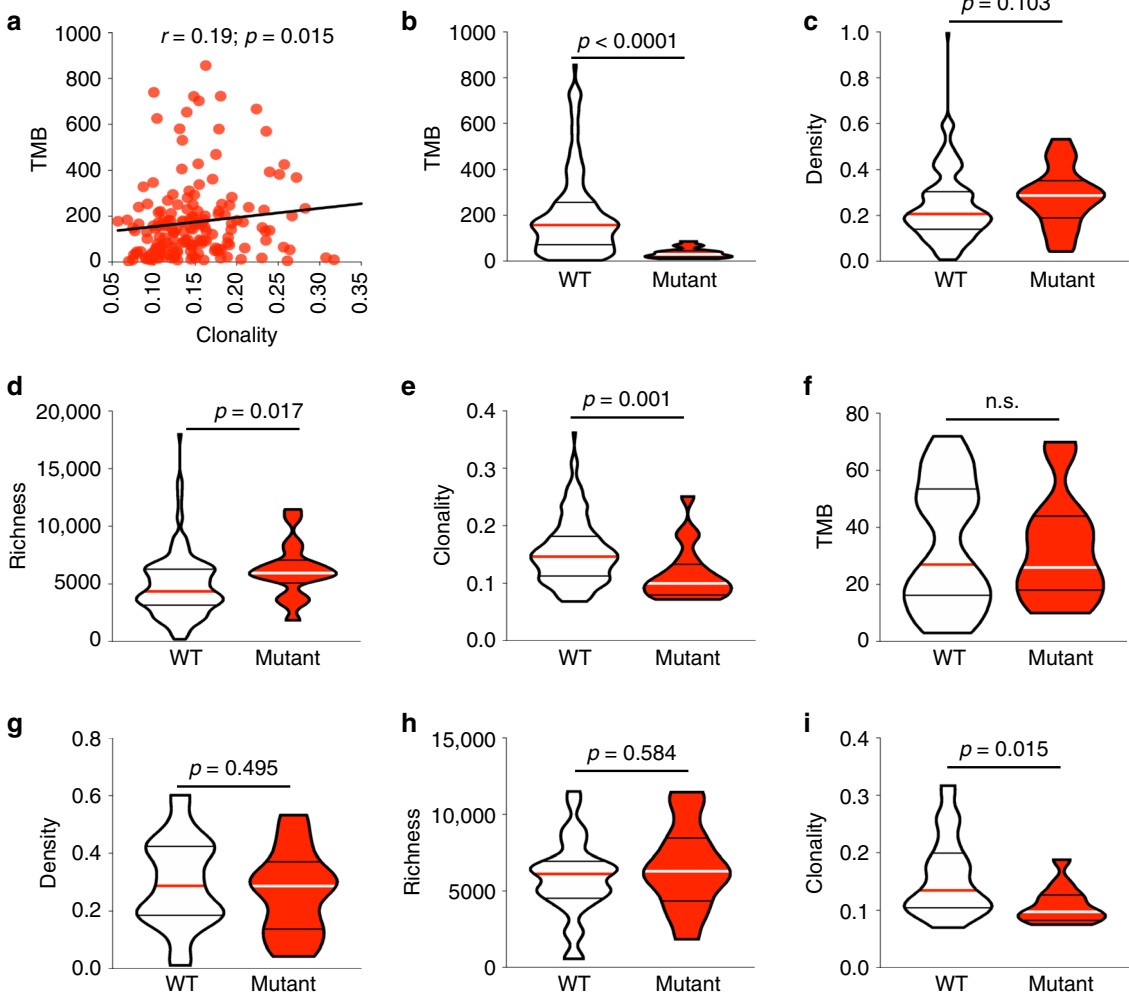

**Fig. 3 T cell clonality is lower in *EGFR* mutant tumors. a** Correlation between tumor mutational burden ($n = 215$) by whole exome sequencing and clonality. **b** Tumor mutational burden ($n = 186$), (**c**) T cell density ($n = 186$), (**d**) richness ($n = 186$) and (**e**) clonality ($n = 186$) in *EGFR* wildtype (white) and mutant (red) tumors. **f** Tumor mutational burden ($n = 51$), (**g**) T cell density ($n = 43$), (**h**) richness ($n = 43$) and (**i**) clonality ($n = 43$) in *EGFR* wildtype (white) and mutant (red) tumors when analyzing only tumors with a low (bottom tertile) TMB. Bars represent median and quartiles.

in T cell density or richness was noted ($n = 224$; Supplementary Fig. 9B–C), suggestive of a more antigen-reactive T cell repertoire in smokers, consistent with prior studies[15]. Notably, higher T cell richness in ADCA patients was associated with a lower rate of recurrence ($n = 134$; $p = 0.026$, Mann–Whitney test, Fig. 4f). These data demonstrate the existence of unique T cell repertoire attributes related to clinicopathologic features in NSCLC tumors and their potential impact on patient outcome.

**T cell clonality is highest in the tumor-adjacent lung.** We next evaluated the T cell repertoire systemically by comparing peripheral blood mononuclear cells (PBMCs), uninvolved tumor-adjacent lungs (≥2 cm from tumor margin without atypia assessed by two pathologists independently), and tumors. T cell density showed no correlation across compartments, though matched samples were positively correlated in richness as well as clonality ($n = 121$; Supplementary Fig. 10A–C). Comparison of the T cell repertoire between these compartments demonstrated a significantly higher T cell density in the tumor than the uninvolved tumor-adjacent lung ($n = 225$; Fig. 5a), though richness was highest in PBMC, as expected. Meanwhile, T cell richness was significantly higher in the tumor compared to the uninvolved tumor-adjacent lung ($n = 224$; Fig. 5b). Surprisingly, T cell clonality was highest in the uninvolved tumor-adjacent lung,

suggesting more focused antigenic responses than in the tumor ($n = 225$; Fig. 5c). These findings could reflect bystander T cell reactivity in the adjacent uninvolved lungs as recently described[36], or an accumulation of exhausted tumor-reactive T cells outside the tumor microenvironment.

We then compared the T cell repertoire of tumor-adjacent, COPD, and healthy lung samples and determined that tumor-adjacent and COPD lungs showed a higher T cell density than lungs from organ donors, presumably reflective of the inflammation in these patients ($n = 253$; $p < 0.0001$, Dunn's multiple comparisons test, Fig. 5d). Interestingly, richness was lowest ($n = 250$; $p < 0.0001$, Dunn's multiple comparisons test) while clonality was highest ($n = 253$; $p < 0.0001$, Dunn's multiple comparisons test) in the uninvolved tumor-adjacent lungs of smokers and non-smokers, highlighting a more active antigenic response that could be related to the tumor (Fig. 5e–f).

**Repertoire homology between tumor-adjacent lung and tumor.** We next evaluated the overlap in T cell repertoire between PBMC, uninvolved tumor-adjacent lung and tumor. Limited homology was noted between the PBMC and uninvolved tumor-adjacent lung or tumor using the Jaccard Index (JI) and Morisita Overlap Index (MOI) ($n = 215$; Fig. 6a and $n = 215$; Supplementary Fig. 11A). However, we observed greater homology

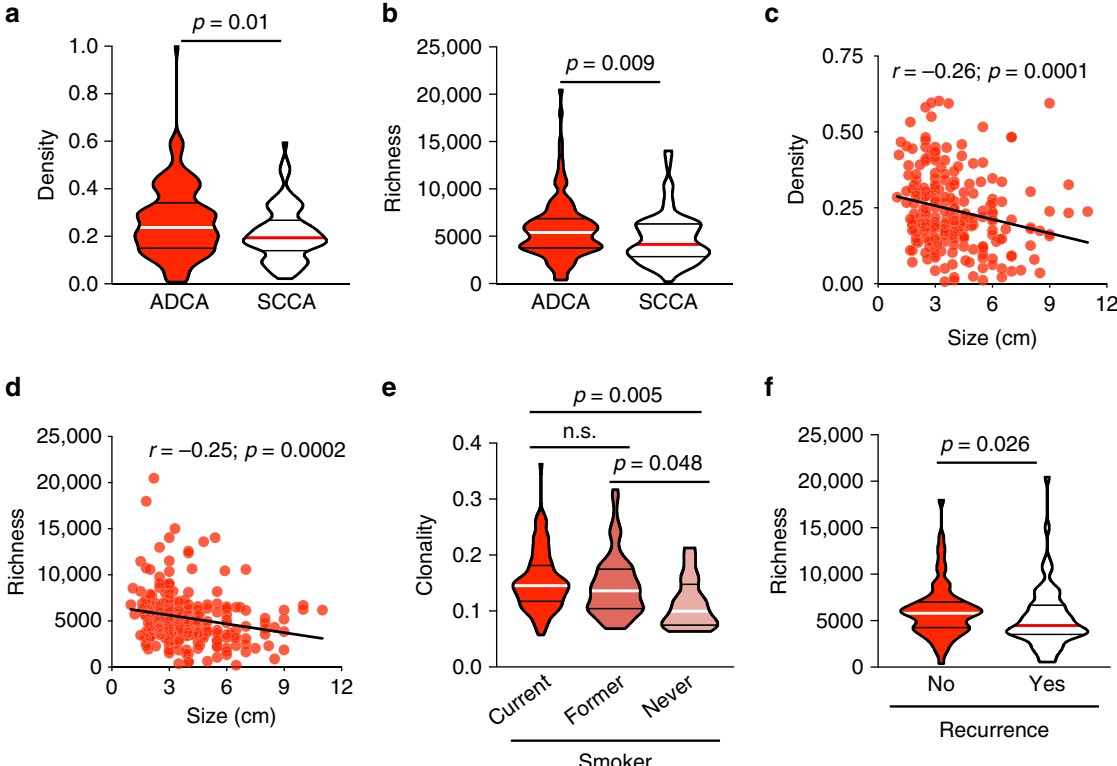

**Fig. 4 The T cell repertoire is associated to clinicopathologic attributes. a** T cell density in adenocarcinomas ($n = 135$) and squamous cell carcinomas ($n = 89$). (**b**) T cell richness in adenocarcinomas ($n = 134$) and squamous cell carcinomas ($n = 89$). **c** Correlation between T cell density or (**d**) richness and tumor size ($n = 225$). **e** T cell clonality in current ($n = 101$), former ($n = 107$), and never smokers ($n = 16$). **f** T cell richness in relapsed ($n = 70$) versus non-relapsed ($n = 64$) adenocarcinoma patients. Bars represent median and quartiles.

between the paired uninvolved tumor-adjacent lung and tumor with both metrics (JI, $p < 0.0001$; MOI, $p < 0.001$, Dunn's multiple comparisons test, Fig. 6a and Supplementary Fig. 11A). Furthermore, among the top 100 most prevalent T cell clones identified in tumors, a median of 57 were detected in uninvolved tumor-adjacent lung tissue, with 28 also among the top 100 most prevalent T cell clones identified in uninvolved tumor-adjacent lungs ($n = 225$; Fig. 6b). Analysis of the lung-enriched T cell repertoire (versus paired PBMC) revealed a 1.9- and 1.8-fold increase in homology between the uninvolved tumor-adjacent lung and tumor by JI ($p < 0.0001$, Dunn's multiple comparisons test) and MOI ($p < 0.0001$, Dunn's multiple comparisons test), respectively, underscoring the parallels in ongoing localized antigenic responses ($n = 215$; Fig. 6c and Supplementary Fig. 11B). Homology across patients was further analyzed, and confirmed hundreds of thousands of CDR3 sequences to be shared (i.e. public TCRs - Supplementary Fig. 12A–D and Supplementary Data 1A–D).

**Shared T cells may target shared mutations or viruses.** As somatic mutations that alter protein sequences can be presented to T cells as neoantigens, we next investigated whether shared nonsynonymous exonic mutations (NSEM) contribute to the T cell repertoire homology between the uninvolved tumor-adjacent lung and tumor. Whole exome sequencing from a subset of 96 patients with available germline DNA from PBMC, paired tumor and uninvolved tumor-adjacent lung tissues demonstrated that an average of only 0.7% of NSEM (0% to 4.5%) were shared between tumor and uninvolved tumor-adjacent lung tissues ($n = 96$; Fig. 7a–e). In regards to the T cell repertoire, a higher proportion of mutations unique to the tumor was modestly associated with a

higher T cell clonality in the tumor ($n = 96$; $r = 0.22$; $p = 0.028$, Spearman rank correlation, Supplementary Fig. 13A), while more unique mutations in the uninvolved tumor-adjacent lung ($r = -0.23$; $p = 0.027$, Spearman rank correlation) or more shared mutations ($r = -0.20$; $p = 0.048$, Spearman rank correlation) was associated with lower tumor T cell clonality ($n = 96$; Supplementary Fig. 13B–C). Though few mutations were shared between the uninvolved tumor-adjacent lung and tumor, a weak but positive correlation was observed between the proportion of shared NSEM and the proportion of shared prevalent T cells suggesting some of the overlap in T cell repertoire may be driven by reactivity to shared mutations/neoantigens ($n = 92$; $r = 0.23$, $p = 0.028$, Spearman rank correlation, Fig. 8a).

Alternately, T cells in the lung could also be targeting viruses. Accordingly, we studied TCR motifs and their antigenic specificity using the GLIPH algorithm[37], a computational tool validated on tuberculosis antigens utilized to predict antigen binding based on comparison of TCR sequencing data to tetramer-validated sequences to identify shared amino acid motifs and infer antigen specificity. To allow comparison of viral and non-viral motifs in spite of the skewing of the database towards non-viral motifs, we normalized the number of non-viral and viral motifs separately, based on whether they were only in the tumor, only in the uninvolved tumor-adjacent lung, or shared in both the tumor and uninvolved tumor-adjacent lung, and proportions were compared. Although the proportion of non-viral-associated TCRs found in the tumor or uninvolved tumor-adjacent lung tissues was generally greater than those from viral TCRs, the T cells shared between tumor and uninvolved tumor-adjacent lung showed a substantial enrichment for predicted viral-associated TCRs in 67% of patients, a 2.1-fold enrichment in the proportion of predicted

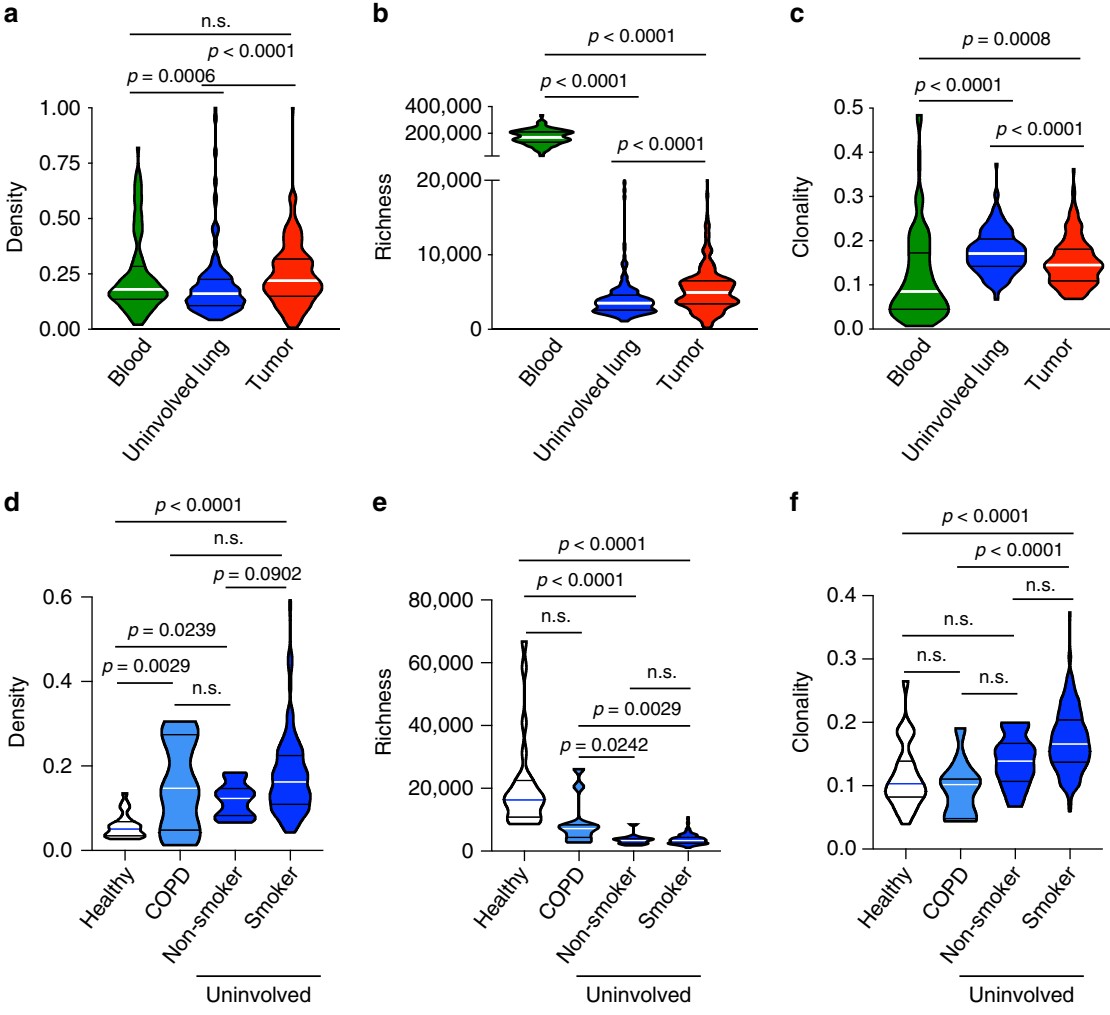

**Fig. 5 T cell clonality is increased in the tumor-adjacent lung. a** T cell density, (**b**) richness, and (**c**) clonality in the peripheral blood (green, $n = 121$), uninvolved tumor-adjacent lung (blue, $n = 216$) and tumor (red, $n = 225$). **d** T cell density ($n = 253$), (**e**) richness ($n = 253$), and **f** clonality ($n = 253$) in healthy (white), COPD (light blue), and tumor-adjacent uninvolved lungs (blue) from smokers and non-smokers. Bars represent median and quartiles.

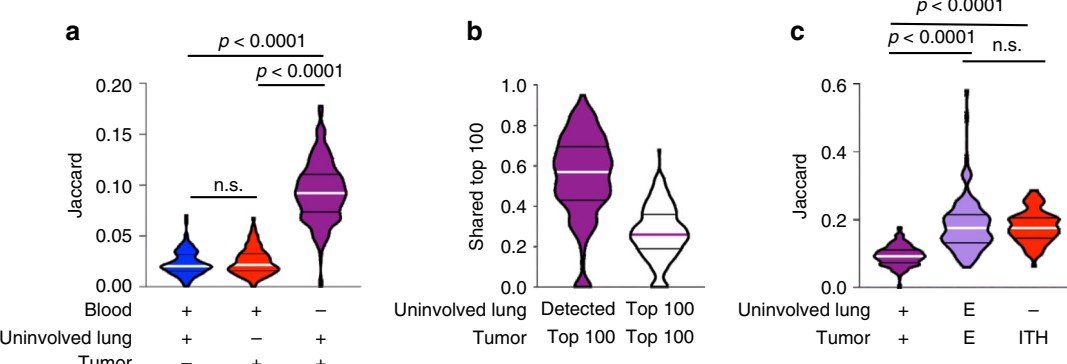

**Fig. 6 T cell repertoire overlap between adjacent uninvolved lung and tumor. a** Jaccard Index when comparing PBMC, uninvolved tumor-adjacent lung, and tumor T cell repertoires ($n = 215$). **b** Proportion of the top 100 T cells in the tumor shared with the uninvolved tumor-adjacent lung ($n = 225$). **c** Jaccard Index in the T cell repertoire when comparing the tumor-adjacent uninvolved lung to tumors, lung-enriched (**e**) T cell repertoire between the tumor-adjacent uninvolved lung and tumor, and different regions of the same tumor (ITH) ($n = 215$). Bars represent median and quartiles.

viral-associated motifs within this group ($n = 178$; $p < 0.0001$; Dunn's multiple comparisons test, Fig. 8b–d). These results suggest that anti-viral T cell responses across the lung may have contributed to T cell repertoire homology between tumor and uninvolved tumor-adjacent lung tissue. Interestingly, a greater proportion of predicted viral motifs were seen in lungs from healthy donors, implying that a larger proportion of the T cell repertoire may be linked to smoking-related inflammation within the lungs of COPD or NSCLC patients ($n = 215$; Fig. 8e–f).

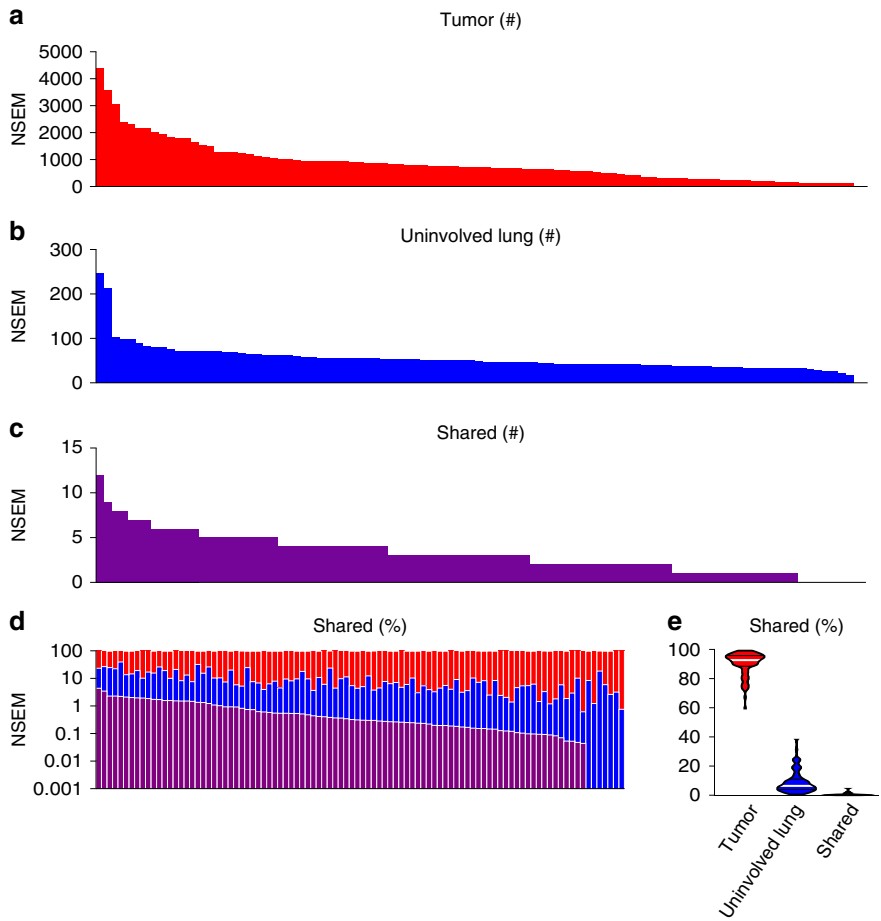

**Fig. 7 Some mutations are shared between the tumor and adjacent uninvolved lung. a** Tumor mutational burden in patients with NSCLC ($n = 96$).
**b** Mutational burden in the tumor-adjacent uninvolved lung ($n = 96$). **c** Number of mutations shared between the tumor and tumor-adjacent uninvolved lung ($n = 96$). **d–e** Proportion of mutations found exclusively in the tumor (red), in the tumor-adjacent uninvolved lung (blue), or shared (purple) ($n = 96$).

**Tumor-focused T cell repertoire linked to better survival.**
Considering the role of the T cell repertoire in anti-tumor responses, we next evaluated its relationship with overall survival (OS) and the results demonstrated that patients with a greater T cell density in peripheral blood had significantly longer OS ($n = 120$; $p = 0.041$, HR: 0.38-0.98, Mantel–Cox test, Fig. 9a). Conversely, a higher T cell density ($n = 216$; $p = 0.036$, HR: 1.024–2.001, Mantel–Cox test, Fig. 9b) and higher T cell clonality in uninvolved tumor-adjacent lung (based on T cells enriched compared to the tumor) correlated with significantly shorter OS ($n = 214$; $p = 0.014$, HR: 1.09–2.138, Mantel–Cox test, Fig. 9c). Multivariate analysis confirmed these associations, with T cell density in the blood ($p = 0.032$, Mantel–Cox test) and clonality in the uninvolved tumor-adjacent lung ($p = 0.032$, Mantel–Cox test) remaining statistically significant, but T cell density in the uninvolved tumor-adjacent lung no longer statistically significant ($p = 0.073$, Mantel–Cox test). Analysis of lung cancer-specific survival revealed much the same trends though smaller numbers may have limited statistical significance ($n = 90$, $n = 157$, and $n = 156$; $p = 0.0717$, $p = 0.1428$, and $p = 0.0511$, respectively; Mantel-Cox test, Fig. 9d–f). As mentioned above, a higher proportion of tumor-only mutations was moderately associated with a higher T cell clonality in the tumor ($r = 0.22$; $p = 0.028$, Spearman rank correlation, Supplementary Fig. 13A), while more mutations unique to the uninvolved tumor-adjacent lung ($r = -0.23$; $p = 0.027$) or more shared mutations ($r = -0.20$; $p = 0.048$, Spearman rank correlation) was associated with lower tumor T cell clonality (Supplementary Fig. 13B–C). Interestingly,

compared to non-relapsed patients, relapsed patients demonstrated a greater proportion of shared mutations and higher level of TCR overlap between the tumor and uninvolved tumor-adjacent lung ($n = 96$ and $n = 215$; $p = 0.011$ and $p = 0.06$, respectively, Mann–Whitney test, Supplementary Fig. 14A–B). Overall, these findings suggest that the host's capacity to generate a stronger T cell response (as indicated by more T cells in PBMC) and a lower density and reactivity of T cells outside the tumor in the uninvolved tumor-adjacent lung (i.e., bystander T cells) may be associated with better survival, while T cell responses targeting viral infections or shared mutations could hamper the immune system's ability to effectively combat the tumor.

## Discussion

Our results highlight the systemic heterogeneity in the T cell repertoire in NSCLC tumors of different histological subtypes and clinicopathological traits, between matched PBMC, uninvolved tumor-adjacent lung and tumor. Exposure to the outside environment complicates T cell analysis in lung tumors, as anti-tumor T cell responses may be intermingled with responses to pathogens and other pro-inflammatory agents. As such, the substantial overlap in the T cell repertoire between the uninvolved tumor-adjacent lung and tumor suggests many T cells may be responding to common antigens throughout the lung. Furthermore, the significantly higher clonality in uninvolved tumor-adjacent lung tissue compared to that of COPD patients and healthy lung donors suggests a more active T cell response, presumably related to anti-tumor surveillance in lung cancer patients

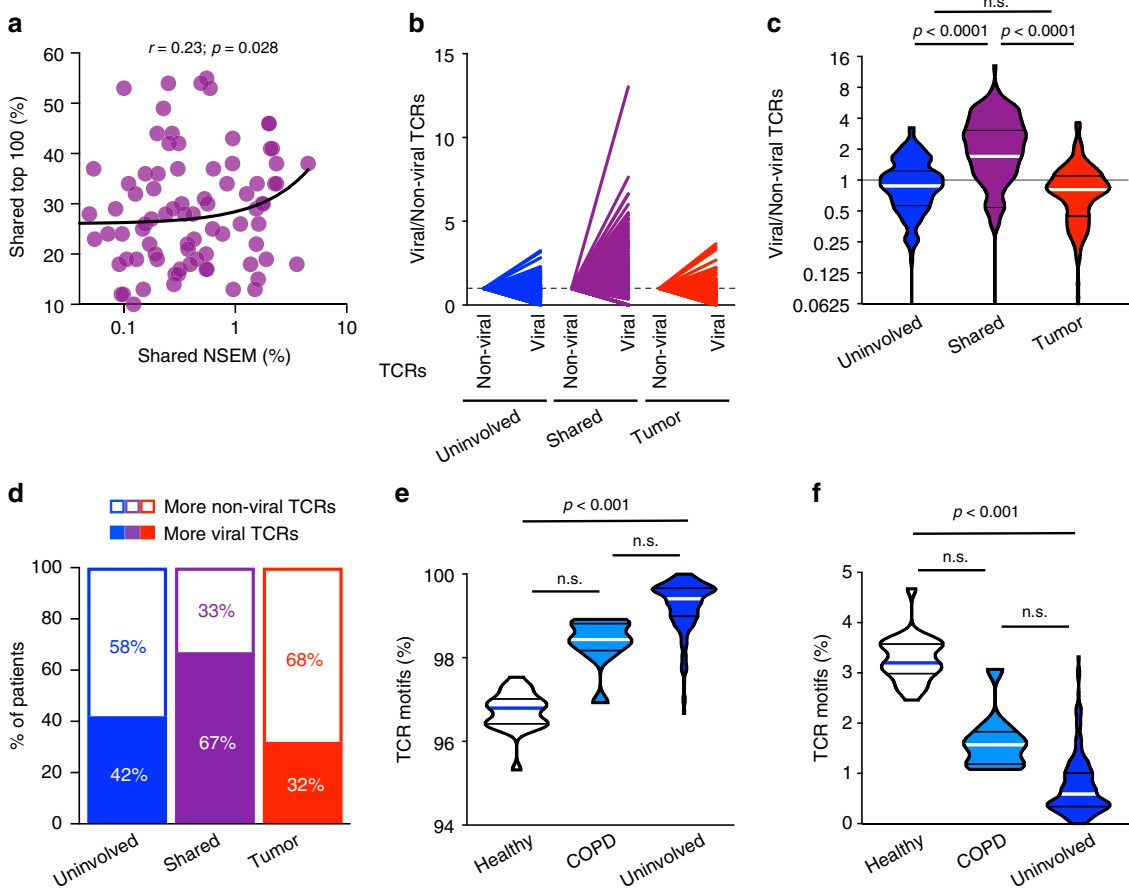

**Fig. 8 Shared T cells may target shared mutations and viruses. a** Correlation between the proportion of shared mutations (NSEM) and shared top 100 T cells between the uninvolved tumor-adjacent lung and tumor ($n = 92$). **b–c** Fold difference between non-viral and viral T cell motifs found exclusively in the tumor-adjacent uninvolved lung (blue), tumor (red), or shared ($n = 178$). **d** Proportion of patients with viral (solid) or non-viral (white) motifs enriched in the uninvolved tumor-adjacent lung (blue), tumor (red), or shared T cells (purple) ($n = 178$). **e** Proportion of non-viral ($n = 215$) and (**f**) viral motifs ($n = 215$) in healthy, COPD, or tumor-adjacent uninvolved lungs. Bars represent median and quartiles.

and/or to promoting factors within the tumor immune microenvironment. A salient finding was the lower clonality observed in NSCLC tumors compared to paired uninvolved tumor-adjacent lungs, indicative of an impaired antigenic response, potentially as a result of an immunosuppressive microenvironment within the tumor, as described by others[14,15].

Though few mutations were detected in the lung, these may play a significant role. The lung as a whole is exposed to the same mutagens, though intact DNA damage repair pathways may lead to the repair of many of these aberrations. However, though less numerous, as many as 247 NSEM were detected in the uninvolved tumor-adjacent lung. Interestingly, the majority of mutations detected in the uninvolved tumor-adjacent lung tissues were not present in the paired tumors and could therefore detract from the host immune response against tumor cells. This is highlighted by the inverse correlation between the number of unique mutations detected in the uninvolved tumor-adjacent lung and the T cell clonality observed within the tumor microenvironment. Interestingly, though few of these mutations were shared between the uninvolved tumor-adjacent lung and tumor, they were found to be increased in patients who relapsed compared to those who did not, and were correlated with the proportion of high frequency T cells shared between the uninvolved tumor-adjacent lung and tumor, which may suggest they play an important role. Furthermore, patients in whom more mutations were shared or unique to the uninvolved tumor-adjacent lung presented lower T cell clonality in their tumors, suggesting these non-tumor-specific

background mutations may be detrimental to the anti-tumor T cell response, and they may blunt the host immune system's ability to clear the tumor, though additional studies are required to validate this hypothesis.

Across numerous tumor types and therapies[11,12,38,39], higher T cell clonality in tumors has been reported in patients with improved clinical benefit, though lack of paired tissues in these studies has prevented analysis of the relationship between uninvolved tumor-adjacent lung and the tumor. However, the exposure of the lung to pathogens highlights the potential for strong antigenic responses unrelated to the tumor within the uninvolved tumor-adjacent lung as well as NSCLC tumors. Importantly, the immune microenvironment likely also plays a role, as immunosuppression in the tumor[14,15] may impede the ability of T cells to expand in response to antigen, thereby preventing the associated increase in clonality.

Finally, our multi-pronged approach highlights the importance of evaluating the relationship between T cell compartments to control for inter-patient and inter-tissue variability. Our findings demonstrate the association between higher T cell density in the blood and improved outcome following surgery, suggesting that the peripheral T cell repertoire in these patients may be reflective of increased systemic immunity. However, the substantial shared T cell population between the matched uninvolved tumor-adjacent lung and tumor may pose therapeutic concerns. TIL-based immunotherapy has been tested in other cancer types[4,40] and has recently become a cause for excitement for lung cancer

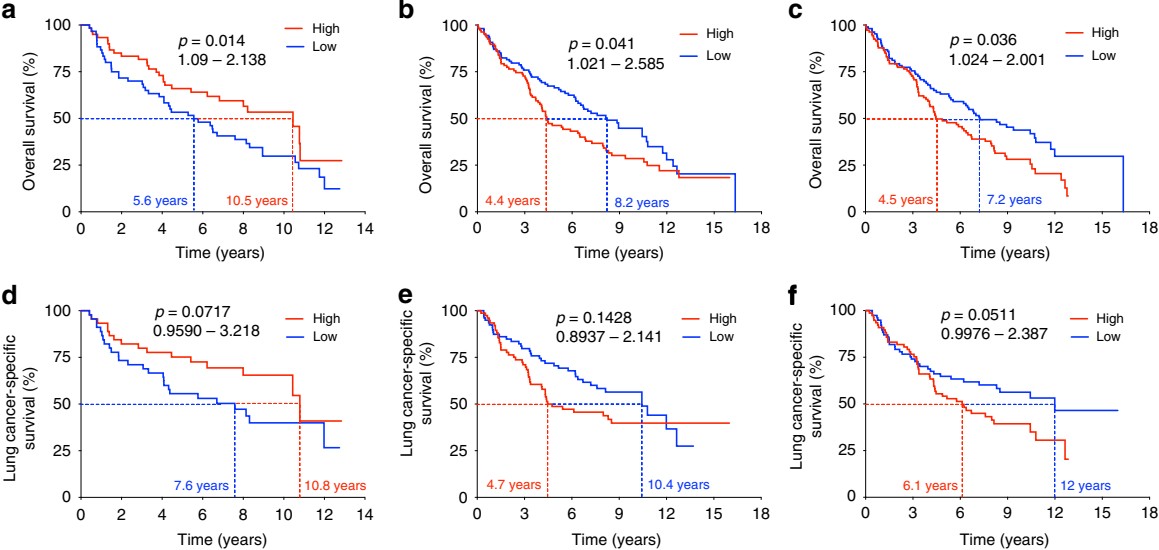

**Fig. 9 A more tumor-focused lung T cell repertoire is associated with improved overall survival (OS). a** Association between high (red) and low (blue) T cell density in PBMC and OS (n = 120). **b** Association between high (red) and low (blue) T cell density in the uninvolved tumor-adjacent lung and OS (n = 216). **c** Association between high (red) and low (blue) T cell clonality in the uninvolved tumor-adjacent lung (enriched compared to the tumor) and OS (n = 214). **d** Association between high (red) and low (blue) T cell density in PBMC and lung cancer-specific survival (n = 90). **e** Association between high (red) and low (blue) T cell density in the uninvolved tumor-adjacent lung and lung cancer-specific survival (n = 157). **f** Association between high (red) and low (blue) T cell clonality in the uninvolved tumor-adjacent lung (enriched compared to the tumor) and lung cancer-specific survival (n = 156). High, above median; Low, below median.

immune therapy (NCT03419559, NCT03215810, NCT02133196). However, TIL expansion and activation is agnostic of antigen restriction. Therefore, because of the substantial overlap of prevalent and potentially reactive T cells between uninvolved tumor-adjacent lung and tumor tissues from NSCLC patients, T cell expansion and ex vivo activation based on their sole presence within NSCLC tumors could result in underwhelming anti-tumor responses or increased immune-related adverse events through expansion of T cell subsets unrelated to the tumor and should thus be taken into consideration in the design and execution of TIL-based therapeutic trials.

## Methods

**Patient cohort and sample collection.** Informed consent was obtained from all study participants. Patient samples were collected as part of the Profiling of Resistance patterns and Oncogenic Signaling Pathways in Evaluation of Cancers of the Thorax study (PROSPECT - LAB07-0233) approved by the University of Texas MD Anderson Cancer Center's Institutional Review Board (IRB). Peripheral blood, uninvolved tumor-adjacent lung, and tumor were collected at time of diagnosis from 236 treatment-naïve NSCLC patients and were a mix of fresh-frozen and FFPE[16,17]. As a control, lung samples were obtained from 11 COPD patients and 24 organ donors. Subject characteristics are presented in Table 1. An overview of all available samples and assays is shown in Supplementary Table 1.

**TCR variable beta chain sequencing.** Sequencing of the CDR3 regions of human TCR-β chains was performed using the immunoSEQ® Assay (Adaptive Biotechnologies, Seattle, WA)[10,41,42]. T cell density was calculated by normalizing TCR-β template counts to the total amount of DNA usable for TCR sequencing, where the amount of usable DNA was determined by PCR-amplification and sequencing of housekeeping genes expected to be present in all nucleated cells. Richness, a measure of the number of unique T cell rearrangements, was calculated using the preseqR package by extrapolating to 400,000 templates for PBMCs and 120,000 templates for tissue. Both richness and clonality are designed to normalize for sampling depth (the number of T cells sampled in a repertoire) to allow fair comparison of samples with different numbers of T cells. Clonality was defined as 1-Peilou's evenness[43]. To identify TCRs that were enriched in one tissue compared to another, we applied a differential abundance framework as described previously[44]. Parameters were as follows: minTotal = 5, productiveOnly = True, alpha = 0.1, count = aminoAcid. Statistical analysis was performed in R version 3.2. TCR sequencing data are available through the immuneACCESS platform (10.21417/AR2019NC - https://clients.adaptivebiotech.com/pub/reuben-2019-natcomms). The immunoSEQ assay is for research use only and not for use in diagnostic procedures.

**Grouping of lymphocyte interactions by paratope hotspots.** For identifying T cell specificity groups, Grouping of Lymphocyte Interactions by Paratope Hotspots (GLIPH) was used to cluster CDR3 rearrangements[37]. Briefly, the CDR3 sequences of the TCR-β chain from the uninvolved tumor-adjacent lung, healthy lung, COPD lung, and tumors were used in conjunction with publicly available, tetramer-defined viral CDR3 sequences[45]. Viral motifs are defined as a GLIPH motif composed of at least 3 viral tetramer-derived CDR3 sequences as well as the enrichment for a given V-gene (p < 0.05 by Fisher's exact test).

**Whole exome sequencing.** Whole exome sequencing (WES) was performed on tumors and uninvolved tumor-adjacent lung tissues to determine the tumor mutational landscape using the NimbleGen 2.1M human exome array and 75bp paired-end sequencing on an Illumina HiSeq2000 in a prior study[16,17]. Pre-processed BAM files were then analyzed to detect single nucleotide variants (SNV) and small insertions and deletions (indels) using MuTect[46] and Pindel[47] algorithms, respectively, against virtual normal sequence developed in-house. Variants were annotated and filtered[48]. In addition, DNA from 96 available matched peripheral blood samples was also sequenced as germline DNA control to identify the mutations in the uninvolved tumor-adjacent lung tissues. Blood DNA was analyzed to identify mutations related to clonal hematopoiesis of indeterminate potential (CHIP) based on annotation specified previously[49]. WES data are available in the EGA (EGAS00001004026).

**Human leukocyte antigen loss of heterozygosity analysis.** For Human Leukocyte Antigen Loss Of Heterozygosity (HLALOH) analysis, we first performed HLA typing using PHLAT[50]. For each patient, we merged tumor and normal BAM files and inferred 4-digit HLA types for the major class I HLA genes (HLA-A, HLA-B and HLA-C). To evaluate HLA loss, we used a computational tool, LOHHLA[51] using purity and ploidy information estimated by Sequenza[52]. As stated in the original paper of LOHHLA, we defined a sample as being subject to HLA loss when any of the two alleles of HLA-A, HLA-B or HLA-C showed a copy number < 0.5 with a paired Student's t test p < 0.01.

**RNA microarray.** RNA microarray was performed in a prior study on 141 patients included here[19,20] using the Illumina HumanWG-6 v3.0 expression bead chip. Then an extended robust multi-array analysis (RMA) background correction model[53] was applied to obtain normalized gene expression profiles for individual samples. Gene expression data are available in the GEO repository (GSE42127).

**Immunohistochemistry.** Tumor tissue was fixed in formalin and embedded in paraffin. For immunohistochemical staining, tissue was cut and mounted at a thickness of 4μm per slide. Slides were then stained with CD3 polyclonal (1:100, DAKO), CD4 clone 4B12 (1:80, Leica Biosystems), CD8 clone C8/144B (1:25, Thermo Scientific), PD-L1 clone E1L3N (1:100, Cell Signaling Technology), PD-1

clone EPR4877-2 (1:250, Abcam), CD45RO clone UCHL1 (ready-to-use, Leica Biosystems), FoxP3 clone 206D (1:50, BioLegend), and Granzyme B clone F1 (ready-to-use, Leica Biosystems)[18]. Slides were then stained using diaminobenzidine as chromogen and the Leica Bond Polymer refine detection kit (Leica Biosystems). Slides were then counterstained with hematoxylin and scanned using an Aperio AT2 automated slide scanner (Leica Biosystems). Quantification was performed on $5 \times 1$ mm$^2$ regions per tumor sample within the tumor center and measuring the average density of positive cells per region as a count of positive cells/mm$^2$. For PD-L1, H-score was calculated by multiplying the proportion of positive cells in the sample (0–100%) by the intensity of staining ($1^+$, $2^+$, or $3^+$) to obtain a score ranging between 1 and 300.

**Statistical analysis**. All plots were generated using GraphPad Prism 8.0 (La Jolla, CA). Because not all TCR variables met the normality assumption, a Kruskal-Wallis test (two-sided) was applied for assessing differences among groups. Wilcoxon matched-pairs signed rank tests were used to compare matched samples. Spearman's rank correlation (two-sided) was used to assess monotonic relationships between two continuous variables. For survival analysis, we first performed univariate Cox analysis on individual TCR variables. Then we fit Cox multiple regression on each TCR variable that tested statistically significant in univariate analysis together with clinical and pathological covariates of interest (age, gender, tumor type, stage, smoking status, and tumor size). Multivariate analysis evaluated each TCR variable with clinical factors taken into account. Due to the exploratory nature of the study, unadjusted p-values not accounting for false-discovery rate (FDR) were used to select TCR variables from univariate analysis.

**Reporting summary**. Further information on research design is available in the Nature Research Reporting Summary linked to this article.

## Data availability

WES data are available in the EGA (EGAS00001004026). RNA microarray data are available in the GEO repository (GSE42127). TCR sequencing data are available through the immuneACCESS platform (10.21417/AR2019NC https://clients.adaptivebiotech.com/pub/reuben-2019-natcomms). Data are available to all researchers upon request. The source data underlying Figs. 1–9 and Supplementary Figs. 1–14 are provided as a Source Data file.

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

## Acknowledgements

This study was supported by the MD Anderson Lung Cancer Moon Shot Program, the Cancer Prevention and Research Institute of Texas Multi-Investigator Research Award grant (RP160668), the National Cancer Institute of the National Institute of Health Research Project Grant (R01CA234629-01), the AACR Johnson & Johnson Lung Cancer Innovation Science Grant (18-90-52-ZHAN), the Cancer Prevention and Research Institute of Texas (R120501), the MD Anderson Physician Scientist Program, the Khalifa Scholar Award, The University of Texas (UT) Systems Stars Award (PS100149), the Welch Foundation Robert A. Welch Distinguished University Chair Award (G-0040), a Department of Defense PROSPECT grant (W81XWH-07-1-0306), the UT Lung Specialized Programs of Research Excellence Grant (P50CA70907), the MD Anderson Cancer Center Support Grant (CA016672), MD Anderson Institutional Support for the Center for Translational and Public Health Genomics, the T.J. Martell Foundation, Adaptive Biotechnologies, the Parker Institute for Cancer Immunotherapy, the Howard Hughes Medical Institute and the Kimberley Clark Foundation Award for Scientific Achievement provided by MD Anderson's Odyssey Fellowship Program.

## Author contributions

A.R., Jie.Z., S-H.C. and R.M.G. contributed equally. A.R. and J.Z. wrote the manuscript. Jie.Z., S-H.C., R.M.G., H.L., P.S., V.B. and J.L. performed statistical analyses. A.R., Jie.Z., S-H.C., R.M.G., W-C.L., K.Q., X.L., F.W., J.L., A.J., X.H., C-J.W., A.K.E., E.Y., R.E., S.B., M.V., X.M., X.S. and Jia.Z. analyzed the data. A. R., S-H.C., C.W., Ch.B., A.S., J.A.W., H.K., P.S., H.R., P.H., J.H., P.S., J.A., M.D., I.W., A.F. and J.Z. interpreted the data. A.R., K.Q., J.L., R.C., A.K.E., L.L., C.G., S.T., R.T. and Jia.Z. generated the data. Ca.B., F.K., C-W.C., X.W., Y.Y., T.C., R.H., S.S., C.M., N.K., A.V., B.S., D.G., J.H. and I.W. obtained clinical samples and data. M.D., I.W., A.F. and J.Z. supervised the study. J.F., E.R.P. and I.I.W. performed pathological assessment of samples. All authors edited the manuscript.

## Competing interests

R.M.G., E.Y, R.E., S.B, M.V. and A.S. have employment and equity ownership with Adaptive Biotechnologies. J.A.W. has honoraria from speakers' bureau of Dava Oncology and is an advisory board member for GlaxoSmithKline and Roche/Genentech. H.R. has employment, equity ownership, patents, and royalties with Adaptive Biotechnologies. P.H. serves on the advisory board of Iovance Biotherapeutics, Inc. and Immatics US. J.V.H. is a consultant for AstraZeneca, Abbvie, Boehringer Ingelheim, Bristol-Myers Squibb, Medivation, ARIAD, Synta, Oncomed, Novartis, Genentech, and Calithera Biosciences, holds stock in Cardinal Spine LLC and Bio-Tree, and has received funding from AstraZeneca. P.S. is a consultant for Bristol-Myers Squibb, Jounce Therapeutics, Helsinn, and GlaxoSmithKline as well as a stockholder from Jounce Therapeutics. J.P.A. is a consultant and stockholder for Jounce Therapeutics, receives royalties from Bristol-Myers Squibb, and has intellectual property with Bristol-Myers Squibb and Merck. I.I.W. receives honoraria from Roche/Genentech, Ventana, GlaxoSmithKline, Celgene, Bristol-Myers Squibb, Synta Pharmaceuticals, Boehringer Ingelheim, Medscape, Clovis, Astra-Zeneca, and Pfizer, and research support from Roche/Genentech, Oncoplex, and HGT. All remaining authors report no conflicts of interest.

## Additional information

Alexandre Reuben [1,2,22], Jiexin Zhang [3,22], Shin-Heng Chiou [4,22], Rachel M. Gittelman [5,22], Jun Li[6], Won-Chul Lee [6], Junya Fujimoto[7], Carmen Behrens[1,7], Xiaoke Liu[1], Feng Wang[6], Kelly Quek [1], Chunlin Wang[8], Farrah Kheradmand[9], Runzhe Chen[1], Chi-Wan Chow[7], Heather Lin[3], Chantale Bernatchez[10], Ali Jalali[11], Xin Hu[6], Chang-Jiun Wu[6], Agda Karina Eterovic[12], Edwin Roger Parra [7], Erik Yusko[5], Ryan Emerson[5], Sharon Benzeno[5], Marissa Vignali[5], Xifeng Wu[13], Yuanqing Ye[13], Latasha D. Little[6], Curtis Gumbs[6], Xizeng Mao[10], Xingzhi Song[6], Samantha Tippen[6], Rebecca L. Thornton[6], Tina Cascone[1], Alexandra Snyder[5], Jennifer A. Wargo [2,6], Roy Herbst[14], Stephen Swisher [15], Humam Kadara[7], Cesar Moran[16], Neda Kalhor[16], Jianhua Zhang[6], Paul Scheet[13], Ara A. Vaporciyan[15], Boris Sepesi[15], Don L. Gibbons[1], Harlan Robins[5,17], Patrick Hwu[10], John V. Heymach[1], Padmanee Sharma[18,19], James P. Allison[19],

Veera Baladandayuthapani[20], Jack J. Lee [20], Mark M. Davis [21]*, Ignacio I. Wistuba[1,7]*,
P. Andrew Futreal [6]* & Jianjun Zhang[1,6]*

[1]Department of Thoracic/Head and Neck Medical Oncology, The University of Texas MD Anderson Cancer Center, 1515 Holcombe, Houston, TX 77030, USA. [2]Department of Surgical Oncology, The University of Texas MD Anderson Cancer Center, 1515 Holcombe, Houston, TX 77030, USA. [3]Department of Bioinformatics and Computational Biology, The University of Texas MD Anderson Cancer Center, 1515 Holcombe, Houston, TX 77030, USA. [4]Institute for Immunity, Transplantation, and Infection Operations, Howard Hughes Medical Institute, 450 Serra Mall, Stanford, CA 94305, USA. [5]Adaptive Biotechnologies, 1551 Eastlake Ave East, Seattle, WA 98102, USA. [6]Department of Genomic Medicine, The University of Texas MD Anderson Cancer Center, 1515 Holcombe, Houston, TX 77030, USA. [7]Department of Translational Molecular Pathology, The University of Texas MD Anderson Cancer Center, 1515 Holcombe, Houston, TX 77030, USA. [8]iRepertoire, Inc., 800 Hudson Way, Suite 2304, Huntsville, AL 35806, USA. [9]Baylor College of Medicine, 1 Baylor Plaza, Houston, TX 77030, USA. [10]Department of Melanoma Medical Oncology, The University of Texas MD Anderson Cancer Center, 1515 Holcombe, Houston, TX 77030, USA. [11]Department of Neurosurgery, The University of Texas MD Anderson Cancer Center, 1515 Holcombe, Houston, TX 77030, USA. [12]Department of Systems Biology, The University of Texas MD Anderson Cancer Center, 1515 Holcombe, Houston, TX 77030, USA. [13]Department of Epidemiology, The University of Texas MD Anderson Cancer Center, 1515 Holcombe, Houston, TX 77030, USA. [14]Department of Medical Oncology, 333 Cedar Street, Yale Cancer Center, New Haven, CT 06510, USA. [15]Department of Thoracic and Cardiovascular Surgery, The University of Texas MD Anderson Cancer Center, 1515 Holcombe, Houston, TX 77030, USA. [16]Department of Pathology, The University of Texas MD Anderson Cancer Center, 1515 Holcombe, Houston, TX 77030, USA. [17]Computational Biology Program, Fred Hutchinson Cancer Research Center, 1100 Fairview Ave North, Seattle, WA 98109, USA. [18]Department of Genitourinary Medical Oncology, The University of Texas MD Anderson Cancer Center, 1515 Holcombe, Houston, TX 77030, USA. [19]Department of Immunology, The University of Texas MD Anderson Cancer Center, 1515 Holcombe, Houston, TX 77030, USA. [20]Department of Biostatistics, The University of Texas MD Anderson Cancer Center, 1515 Holcombe, Houston, TX 77030, USA. [21]Department of Microbiology and Immunology, Stanford, 450 Serra Mall, Stanford, CA 94305, USA. [22]These authors contributed equally: Alexandre Reuben, Jiexin Zhang, Shin-Heng Chiou, Rachel M. Gittelman. *email: mmdavis@stanford.edu; iiwistuba@mdanderson.org; afutreal@mdanderson.org; jzhang20@mdanderson.org

