## [Peer Review File · Nature Communications]

Reviewers' comments:

Reviewer #1, expert on T cell repertoire in cancer (Remarks to the Author):

This manuscript by Reuben et al. describes correlative analysis of beta chain TCR sequences found in lung tumor samples plus adjacent matched normal samples and in some cases matching peripheral blood samples. This work is supplemental to previously published analysis of IHC, gene expression and exome sequencing data from the same cohort. Although the study is entirely observational, the new TCR repertoire analysis offers an interesting and informative view of T cell distributions in these samples and some useful insights into T cell immunity in lung cancer.

1. A description is needed of how the repertoire sequence data was normalized to allow meaningful comparisons among samples, in terms of diversity, richness, clonality, etc... There is brief mention of a normalization step in the methods for FFPE samples but it is not interpretable and there is no supporting data. Are there sample types other than FFPE?

2. Unfortunately tumor samples are only a snapshot in time. Therefore, it can't be inferred whether mutation burden was high and then became low due to immune editing (ie. depletion of tumor clones by reactive T cells), or rather that mutation burden was low initially and failed to elicit T cell reactivity. As such it is difficult to interpret the observed relationship between EGFR mutation status, TMB and T cell clonality. This fundamental issue should be addressed by the authors in the revised manuscript.

3. The authors use the term "T cell response" frequently. However, the analysis as presented can't actually distinguish between tumor-reactive T cells and bystander/incidental T cells. All T cells are simply "associated" and in my view the term T cell response should not be used without direct evidence of tumor antigen recognition. Approaches that could help with inferring the presence of responder T cells are i) association with HLA loss variants, ii) viewing the repertoire attributes after splitting samples according to high/low status of IHC markers of antigen experience (CD45RO) and activation (PD-1) and iii) segregating tumor samples according to whether the T cells are denser in stroma versus epithelium. I realize it would be a lot of additional work to do these analyses, but a strength of this study is the extensive orthogonal data available for the cohort - it would be good to fully utilize these existing resources if possible.

4. It's not clear how mutations in matched normal tissue are identified. Ordinarily, bona fide tumor mutations would be identified in comparison to matched normal or to constitutional DNA from peripheral blood. Here, are both tumor and matched normal variants inferred by comparison to DNA from peripheral blood? If so are any measure taken to ensure mutations shared between tumor and matched normal are not actually mutations in peripheral blood DNA? This is not a major criticism, I just think it needs more explanation in the methods. The scenario is perhaps unique to lung cancer, where background mutational load in non-tumor tissue is expected to be very high. Also, on that note, I think it would be better to actually refer to variants in adjacent normal tissue as background mutations, rather than as passenger mutations (eg. lines 49, 302). There can only be passengers if there is a driver, in which case these samples would be tumors, not tumor adjacent normal tissue.

5. What exactly is "a greater T cell density in peripheral blood" and how is this measured? Is this inferred from the TCR sequencing data (if so, how) or is this referring to an absolute lymphocyte count from blood work? If the former, does it correlate with the latter?

6. I'm not sure that "less clonal T cells in tumor-adjacent normal lung" (line 266) can be taken to indicate a more tumor-focused T cell repertoire.

7. An accounting of public TCRs (ie. which TCRs from which sample types are shared among 2 or more patients) would be a useful addition to the paper.

Reviewer #2, expert on clinical cancer immunotherapy and biomarkers of response (Remarks to the Author):

This manuscript describes T cell repertoire analysis for 236 early stage lung cancer tumors and adjacent lung. Their conclusion is that a "considerable proportion" of TCRs "appeared to be reactive to shared passenger mutations or viral infections". Higher sharing of TCRs in tumors and adjacent lung was associated with inferior survival. They indicate that this shows that "a concise understanding of shared antigens and T cells between tumor and adjacent normal tissue in NSCLC is needed to improve therapeutic efficacy and reduce risk of toxicity in the context of immunotherapy, particularly adoptive T cell therapy". The numerous enumerations and correlations of TCR features with each other and speculations as to their significance in this population are of modest interest, and there are other concerns.

Major:

1. The tumors were from resections from patients not treated with immunotherapy, which makes all conclusions relating to immunotherapy speculative.
2. The entire study is highly correlational and they seem to confuse correlation with causation throughout the manuscript and appear to use univariate analyses even when evaluating dozens of comparisons.
3. Not much attention is paid to potential regional heterogeneity in TIL within tumors
4. No analysis of tumor differentiation with TIL/TCRs is done.
5. Many of the correlations (such as 4C, and S10 F, G, H are marginally significant and completely unconvincing visually
6. The GLIPH program described in reference 32 was really only carefully tested against tuberculosis antigens, and the "viral T cell responses" argument is also speculative.
7. All of the survival correlations in figure 5 are of marginal statistical significance, and T-cell density has long been shown to be correlated with survival.
8. Also, as the median time to recurrence after surgery is 11 months and the median survival after metastatic relapse is 6 to 8 months, the figures showing overall survival out to 15 years suggests that most of the deaths in this group were not from lung cancer relapse, but rather other causes. Lung cancer specific mortality should be shown.

Minor:

1. Adjacent lung is not "normal", and this word should be removed from the description
2. "NSEM" is not defined in its first use in the figure legend to S4.
3. In S5H it would be more interesting to look at clonality in EGFR mutants with a high TMB rather than low.

Reviewer #3, expert in clinical lung cancer genomics (Remarks to the Author):

The authors thoroughly evaluated T cell repertoire based on TCR sequencing of CDR3 variable regions in 236 resected non-small cell lung cancer (NSCLC) tumors, and their matched tumor-adjacent normal lung as well as peripheral blood. These tumor genomic and microenvironment profiles were reported in Karada et al. (2017). They assessed the associations between T cell repertoire features (diversity and clonality) and tumor genomic, microenvironment, and clinicopathological aspects. They identified that characteristics of T cell repertoire in tumor-adjacent normal tissue were distinguished from non-cancer normal lung and highly associated to nearby tumors. Furthermore, the authors found the patient outcome was associated with T cell clonality in tumor-adjacent normal tissue and density of T cells in PBMC.

This is a well-designed study providing insight into the process of immune editing in early stage of NSCLC, as well as the impact in relation to neighboring tissues. The abnormality of T cell repertoires in the tumor-adjacent normal tissue suggests that future consideration should be focused on understanding and harnessing the activity of increased anti-tumor T cells that have repertoires that are distinct from adjacent normal regions. However, there are several minor points that authors should address to maximize the impact of the study

1. The authors mentioned that the genomic profiles of the cohort were reported in Karada et. al (reference 23). However, Karada et al. reported whole exome sequencing data of 108 tumor and normal pairs. The authors should clearly indicate how many tumor samples in this study have genomic data profiles, and were subjected to the analyses exploring the relationship between T cell repertoire and genomic features (TMB, driver genes).

2. The authors observed that T cell clonality was highly correlated to cytotoxic phenotype (i.e. CD8 T cells, GZMB positive cells) and TMB. However, they also found that TMB was highly associated with GZMB positive cells. Therefore, it is essential to perform multivariate analysis to determine if the number of GZMB positive cells and TMB (in log scale) are two independent factors that determine T cell clonality.

3. The Kruskal-Wallis test has been utilized to assess the differences among groups (i.e. Figure 3). However, the TCR variables (e.g. clonality) are paired (i.e. tumor and matched adjacent normal), and vary over a wide range, it would be more appropriate to utilize a paired test (i.e. Wilcoxon signed-rank test) to confirm the differences between compartment pairs.

4. A more comprehensive approach would be achieved if the analyses of T cell repertoire in the adjacent normal (Figure S8C-E) were based on comparisons of four groups (healthy, COPD, never-smokers and smokers) to distinguish the effects of cancer and smoking to adjacent normal regions.

5. On page 10 and Figure S9B-C, "lung-enriched T cell repertoire" was mentioned. The authors should clarify how this is defined. Are they top 100 T cell clones detected in the specific compartment?

6. The tumor-adjacent normal regions have low T cell infiltration and diversity (Figure 4A-B), but higher clonality (Figure 4C) compared to matched tumors. On page 9 the authors suggest that there are high antigenic responses in tumor-adjacent normal regions. However, T cell clones between tumor and adjacent normal were highly overlapped (Figure 4B). Together these results also imply that specific T cell clones detected in the adjacent normal regions are prevalent tumor-target T cell clones targeting shared mutations in both tumor and adjacent normal tissue. These T cells in the adjacent normal tissues might reflect immune exhaustion and suppression in nearby tumor areas. This could be another explanation for the association between the poor survival and high level of T cell density and clonality in adjacent normal tissues (Figure 5). The findings are very much in line with, and support, the rich literature documenting an immune suppressed tumor microenvironment in NSCLC.

Reviewer #1

This manuscript by Reuben et al. describes correlative analysis of beta chain TCR sequences found in lung tumor samples plus adjacent matched normal samples and in some cases matching peripheral blood samples. This work is supplemental to previously published analysis of IHC, gene expression and exome sequencing data from the same cohort. Although the study is entirely observational, the new TCR repertoire analysis offers an interesting and informative view of T cell distributions in these samples and some useful insights into T cell immunity in lung cancer.

1. A description is needed of how the repertoire sequence data was normalized to allow meaningful comparisons among samples, in terms of diversity, richness, clonality, etc... There is brief mention of a normalization step in the methods for FFPE samples but it is not interpretable and there is no supporting data. Are there sample types other than FFPE?

Response: The reviewer raises an important point regarding the normalization step to ensure comparability across samples. In our study, samples were a mix of FFPE and fresh frozen (FF) though tumor/uninvolved lung pairs were always matched by sample type. Normalization was in fact performed for both FFPE and FF samples. The formulas utilized for calculating richness and clonality are designed to normalize for sampling depth (the number of T cells sampled in a repertoire) regardless of tissue type. In our prior submission *Methods*, we inadvertently omitted any mention of normalization for FF samples, which may have led to confusion. We have now corrected this statement in the *Methods* for TCR sequencing which reads as such (line 399):

"T cell density was calculated by normalizing TCR- β template counts to the total amount of DNA usable for TCR sequencing, where the amount of usable DNA was determined by PCR-amplification and sequencing of housekeeping genes expected to be present in all nucleated cells. Richness, a measure of the number of unique T cell rearrangements, was calculated using the preseqR package by extrapolating to 400,000 templates for PBMCs and 120,000 templates for tissue. Both richness and clonality are designed to normalize for sampling depth (the number of T cells sampled in a repertoire) to allow fair comparison of samples with different numbers of T cells."

2. Unfortunately tumor samples are only a snapshot in time. Therefore, it can't be inferred whether mutation burden was high and then became low due to immune editing (ie. depletion of tumor clones by reactive T cells), or rather that mutation burden was low initially and failed to elicit T cell reactivity. As such it is difficult to interpret the observed relationship between EGFR mutation status, TMB and T cell clonality. This fundamental issue should be addressed by the authors in the revised manuscript.

Response: Reviewer 1 raises a key point regarding the static nature of our analyses and the fact they cannot discriminate with certainty between a lower TMB due to immune editing or simply a lack of immunogenic mutations. This is unfortunately a limitation of our study and of any retrospective cohorts. However, our $EGFR_{MUT}$ data suggest these tumors probably lack immunogenic mutations to begin with. As shown in **Fig. S8**, $EGFR_{MUT}$ tumors exhibit a lower TMB than their wildtype counterparts, consistent with prior studies (Offin *et al.*, CCR, 2019). In order to adjust for this difference in TMB between groups, we selected only $EGFR_{WT}$ tumors within the lowest TMB tertile ($EGFR_{MUT}$ tumors were all included within the lowest TMB tertile). The results showed that although the TMB was comparable between $EGFR_{MUT}$ and $EGFR_{WT}$

TMB^{lo} tumors in this subgroup (**Fig. S8E**), T cell clonality remained higher in *EGFR_{WT}* tumors ($p=0.015$, **Fig. S8H**). These results suggest that 1) *EGFR_{WT}* tumors could potentially induce better T cell expansion regardless of a low TMB or 2) alternatively that the low TMB in these tumors may have resulted from depletion of immunogenic tumor clones by reactive T cells, and as a result driven down the TMB as suggested by *Reviewer 1*. Conversely, clonality was consistently lower in *EGFR_{MUT}* tumors, thereby suggesting that although these tumors have an identical TMB to *EGFR_{WT}* TMB^{lo} tumors, T cells may not be expanding (leading to low clonality). This could suggest that in *EGFR_{MUT}* tumors 1) there was a lack of immunogenic mutations, or 2) there was existence of alternative immunosuppressive mechanisms, which prevent antigen recognition and T cell expansion. Though we believe our data support this hypothesis, it does remain speculative as highlighted by *Reviewer 1* and we cannot exclude the possibility that 1) *EGFR_{MUT}* tumors experienced immune editing and samples analyzed were collected *following* this response, and 2) that *EGFR_{WT}* tumors exhibited T cell expansion due to factors other than immunogenic mutations, such as exposure to cytokines in the tumor microenvironment. Therefore, we have revised the language in order avoid overstating this conclusion and to highlight the need for additional studies. The *Results* now read as such (line 172):

“In order to adjust for this difference in TMB, we focused exclusively on EGFR_{WT} tumors within the lowest TMB tertile. Although TMB was comparable between EGFR_{MUT} and EGFR_{WT} TMB^{lo} tumors in this subgroup (Fig. S8E), T cell clonality remained higher in EGFR_{WT} tumors ($p=0.015$, Fig. S8F-H). These results suggest that EGFR_{WT} tumors could potentially induce better T cell expansion regardless of a low TMB or alternatively that the low TMB in these tumors may have resulted from depletion of immunogenic tumor clones (more likely with higher TMB) by reactive T cells, and as a result driven down the TMB. Conversely, clonality was consistently lower in EGFR_{MUT} tumors, thereby suggesting that T cells may not be expanding (leading to low clonality), most likely due to the existence of alternative immunosuppressive mechanisms, which prevent antigen recognition and T cell expansion. Of note, even within the highest TMB EGFR_{MUT} tumors, no differences were observed in T cell repertoire attributes (Fig. S9A-F). Taken together, these results suggest that there exist TMB-independent mechanisms contributing to the low clonality in EGFR_{MUT} NSCLC tumors. Otherwise, no associations were observed between the T cell repertoire and other frequently mutated cancer genes in NSCLC such as KRAS and TP53.”

Furthermore, in response to *Reviewer 2* (question #18 below), we have performed analyses demonstrating that even within the highest TMB *EGFR_{MUT}* tumors, there are no differences in T cell repertoire attributes, whether it be density, richness or clonality. These results are shown below for convenience (**Reviewer Fig. RF1**) and further support the lack of a role for the TMB

in driving T cell responses in $EGFR_{MUT}$ tumors. This figure is also now included in our manuscript as **Fig. S9**.

Reviewer Figure RF1. T cell repertoire features are not associated with higher tumor mutational burden among $EGFR$ -mutant patients. (A-C) Comparison of TMB^{hi} (above median, white) or TMB^{lo} (below median, red) tumors relative to A) T cell density, B) T cell richness, and C) T cell clonality. Correlation between (D-F) tumor mutational burden and D) T cell density, E) richness, and F) clonality only in tumors harboring classical $EGFR$ mutations.

3. The authors use the term "T cell response" frequently. However, the analysis as presented can't actually distinguish between tumor-reactive T cells and bystander/incidental T cells. All T cells are simply "associated" and in my view the term T cell response should not be used without direct evidence of tumor antigen recognition. Approaches that could help with inferring the presence of responder T cells are i) association with HLA loss variants, ii) viewing the repertoire attributes after splitting samples according to high/low status of IHC markers of antigen experience (CD45RO) and activation (PD-1) and iii) segregating tumor samples according to whether the T cells are denser in stroma versus epithelium. I realize it would be a lot of additional work to do these analyses, but a strength of this study is the extensive orthogonal data available for the cohort - it would be good to fully utilize these existing resources if possible.

Response: We understand *Reviewer 1*'s concern as to the use of the term "response" in absence of functional evidence thereof. We further agree with the hypothesis that the T cells in the lungs could be bystander/incidental T cells as this is an important hypothesis highlighted in our study and others. In order to further elucidate whether T cells were potentially responding to tumor antigens, we have now completed all 3 analyses suggested by *Reviewer 1*, which we now include in the revised manuscript.

First, analysis of HLALOH was performed using available whole exome and RNA expression data. Patients exhibiting HLALOH were compared to those without HLALOH as to T cell repertoire attributes in the tumor. The results are presented in **Reviewer Fig. RF2** (and **Fig. S7**) and demonstrate no clear association between HLALOH and T cell density, richness, or clonality suggesting this may not represent a major active mechanism of resistance in our cohort and that the T cells present in the tumor may in fact be bystander T cells, as supported by other parts of our manuscript.

Reviewer Figure RF2. No association between HLA loss of heterozygosity and T cell repertoire attributes. Comparison of **A)** T cell density, **B)** T cell richness, and **C)** T cell clonality in tumors exhibiting no HLA loss of heterozygosity (HLALOH, Red) and those exhibiting HLALOH (White).

We have also performed analyses based on dichotomizing expression of CD45RO and PD-1 into high (above median) and low (below median). Results are shown in **Reviewer Fig. RF3** (and **Fig. S4**) and demonstrate a higher T cell density in tumors with high CD45RO expression ($p=0.0016$, **Reviewer Fig. RF3A**), as well as a trend towards increased T cell richness in the same group ($p=0.0851$, **Reviewer Fig. RF3B**) though no difference was seen with clonality ($p=0.5027$, **Reviewer Fig. RF3C**). This suggests that T cell density and diversity may be driven by an influx of antigen-experienced T cells (CD45RO+) in the tumor, perhaps due to the increase in chemokine receptors allowing homing to sites of inflammation which occurs as a result of antigen-exposure. However, the lack of association with clonality suggests these cells may not be expanding and may not actively contribute to the anti-tumor response in this cohort, though this requires validation. This may also suggest antigen-experienced cells in the tumor microenvironment may partially be represented by bystander T cells.

With regards to PD-1 expression, PD-1^{hi} tumors were associated with higher T cell density ($p<0.0001$, **Reviewer Fig. RF3D**), richness ($p=0.0083$, **Reviewer Fig. RF3E**), and clonality ($p=0.0104$, **Reviewer Fig. RF3F**), which could highlight both the recruitment of these cells to the tumor microenvironment and their ability to expand, thereby increasing clonality. This is in line with PD-1 having served as a marker of activation in numerous studies to date and suggests many T cells in the tumor may be reactive against tumor antigens (Gros *et al.*, Nature Medicine, 2016).

Reviewer Figure RF3. CD45RO and PD-1 expression are associated with T cell repertoire attributes. T cell density (A and D), richness (B and E) and clonality (C and F) in tumors with high (above median) or low (below median) CD45RO density (A-C) or PD-1 density (D-F) by immunohistochemistry.

Finally, in regards to the analysis of tumor and stroma, we performed assessment of IHC samples to quantify CD3, CD4 and CD8 T cell density at the tumor center and periphery. We then calculated a ratio of CD3, CD4, and CD8 T cells at the periphery and center to determine the gradient of T cell enrichment, whether T cells may be in any way excluded from tumors (peripheral density > center density) and how this could impact attributes of the T cell repertoire. As shown in **Reviewer Fig. RF4** (and **Fig. S5**), no differences are seen between CD3 or CD8 enrichment at the tumor center or periphery and T cell density (CD3, $p=0.1473$; CD8, $p=0.2394$), richness (CD3, $p=0.1324$; CD8, $p=0.4699$), and clonality (CD3, $p=0.1915$; CD8, $p=0.2132$). However, with CD4, though no statistical difference was seen in T cell density ($p=0.0901$), T cell richness ($p=0.0006$) was significantly higher while clonality is significantly lower ($p=0.0120$) when CD4 was enriched at tumor center. These findings could suggest different things. First, they suggest CD4 T cells may be the predominant population of T cells recruited to the tumor. This is supported by prior work by our group and others demonstrating the majority of lung tumors are enriched for CD4 rather than CD8 T cells (**Fig. S2B-C** and Reuben *et al.* Cancer Discovery, 2017). Secondly, CD4 T cells have the potential to inhibit T cell expansion and proliferation if they are regulatory (Treg). This means that an increased frequency of CD4 T cells at the tumor center could suppress CD8 T cell responses locally. Similar analysis performed on FoxP3 (data not shown), a marker associated with Tregs also transiently upregulated by activated CD8 T cells, demonstrated similar trends to those observed in CD4, though statistical significance was not attained, with higher richness ($p=0.0920$) and lower clonality ($p=0.1296$) in tumors with a higher FoxP3 density at the center. This supports the hypothesis that most CD4 in the tumor may have been Tregs.

Reviewer Figure RF4. T cell density, richness and clonality in tumors enriched for T cells at the tumor center or periphery. T cell density (A, D, G), richness (B, E, H) and clonality (C, F, I) in tumors with a higher density of CD3 (A-C), CD4 (D-F) or CD8 (G-I) in their center (white) or periphery (red).

4. It's not clear how mutations in matched normal tissue are identified. Ordinarily, bona fide tumor mutations would be identified in comparison to matched normal or to constitutional DNA from peripheral blood. Here, are both tumor and matched normal variants inferred by comparison to DNA from peripheral blood? If so are any measures taken to ensure mutations shared between tumor and matched normal are not actually mutations in peripheral blood DNA? This is not a major criticism, I just think it needs more explanation in the methods. The scenario is perhaps unique to lung cancer, where background mutational load in non-tumor tissue is expected to be very high. Also, on that note, I think it would be better to actually refer to variants in adjacent normal tissue as background mutations, rather than as passenger mutations (eg. lines 49, 302). There can only be passengers if there is a driver, in which case these samples would be tumors, not tumor adjacent normal tissue.

Response: *Reviewer 1* brings up an interesting issue, which had not been considered in our initial analysis, which is the possibility that mutations detected in the uninvolved lung and tumor are actually mutations found in the peripheral blood samples used as a reference, recently referred to as CHIP mutations (Clonal Hematopoiesis of Indeterminate Potential). These mutations have been found in blood or bone marrow cells from healthy subjects without leukemia, but can then lead to the eventual development of leukemia. In order to assess this possibility, we identified mutations related to CHIP in blood samples (Jaiswal *et al.* NEJM - 2014). As shown in **Reviewer Table RT1**, only 8 such mutations were detected in our samples, and all were present at low allele frequencies. This means that the wildtype variants would have been selected as the germline control when comparing to the uninvolved lung and tumor DNA and these mutations in the blood are therefore not to blame for the shared mutations seen between the uninvolved lung and tumors. Indeed, we checked these loci in corresponding tumor samples and uninvolved normal lung samples from patients who had these CHIP mutations detected in the blood but did not find any mutations in tumor samples and uninvolved normal lung samples resulted from these potential CHIP mutations.

Reviewer Table RT1. List of CHIP mutations identified.

Chromosome	Start	Context	Ref	Alt	VAF	Function	Gene	Function
X	123184969	TTTxGCA	A	T	0.24	splicing	STAG2	NA
17	7578190	TCAxAGG	T	C	0.04	exonic	TP53	nonsyn SNV
20	57484421	GCCxTGT	G	A	0.24	exonic	GNAS	nonsyn SNV
9	5073770	TGTxTCT	G	T	0.17	exonic	JAK2	nonsyn SNV
2	25467478	TGGxAGC	T	C	0.06	exonic	DNMT3A	nonsyn SNV
17	7577548	TGCxGCC	C	A	0.07	exonic	TP53	nonsyn SNV
17	7577568	TTAxACA	C	T	0.29	exonic	TP53	nonsyn SNV
9	5073770	TGTxTCT	G	T	0.23	exonic	JAK2	nonsyn SNV

We thank *Reviewer 1* for pointing out this possibility. To clarify how these analyses were performed, we have added language to the *Methods* section, which reads as such (line 427):

"In addition, DNA from 96 available matched peripheral blood samples was also sequenced as germline DNA control to identify the mutations in the uninvolved tumor-adjacent lung tissues. Blood DNA was analyzed to identify mutations related to clonal hematopoiesis of indeterminate potential (CHIP) based on annotation specified previously.

Furthermore, as suggested, we have changed all mentions of "passenger mutations" to "background mutations" in the manuscript text.

5. What exactly is "a greater T cell density in peripheral blood" and how is this measured? Is this inferred from the TCR sequencing data (if so, how) or is this referring to an absolute lymphocyte count from blood work? If the former, does it correlate with the latter?

Response: We apologize for the lack of clarity in our initial description of the T cell density calculation from peripheral blood. T cell density was inferred from TCR sequencing as the number of nucleated cells input into the assay via DNA mass input (absorbance). The number of T cells is precisely quantified from the sequencing data using Adaptive Biotechnologies' synthetic control repertoire that allows the conversion of sequencing reads into template

molecule counts. The density estimate is then simply the number of T cells divided by the number of total nucleated cells. We have added this information to the manuscript in the *Methods* section for clarity, which now reads as such (line 399):

"T cell density was calculated by normalizing TCR-β template counts to the total amount of DNA usable for TCR sequencing, where the amount of usable DNA was determined by PCR-amplification and sequencing of housekeeping genes expected to be present in all nucleated cells."

In regards to absolute lymphocyte counts, though these were not part of our initial analyses, we have now reviewed medical charts for the 120 patients with TCR sequencing on PBMC and retrieved absolute lymphocyte counts and frequencies from routine CBC assays. These counts were then correlated with T cell density inferred from TCR sequencing to study the relationship between both parameters. Though T cell density correlated well with CD3 counts by IHC in tumor tissue ($r=0.54$, $p<0.0001$, **Reviewer Fig. RF5A**), we observed no correlation between T cell fraction by TCR sequencing and by lymphocyte count from blood work both as an absolute number and frequency (**Reviewer Fig. RF5B-E**). This could be due to several factors. First, the same tumor tissue sample was used for TCR sequencing and IHC, limiting any temporal/longitudinal variability. This is unfortunately not the case for lymphocyte counts, where dates were divergent between TCR sequencing and absolute lymphocyte counts. Second, though T cell fraction by TCR sequencing assesses only T cells as a proportion of all nucleated cells, absolute lymphocyte counts from CBC are based on cell size, nuclei, and include both B cells and T cells, which can make them less accurate for T cell assessment. We believe this could explain some of the lack of correlation between both platforms.

Reviewer Figure RF5. Correlation between blood lymphocyte counts and T cell fraction estimated through TCR sequencing. A) Correlation between CD3 IHC and T cell fraction by TCR sequencing in NSCLC tumors. **B-E)** Correlation between T cell fraction by TCR sequencing in the blood and absolute lymphocyte counts (**B-C**) and frequencies (**D-E**) prior to surgery (**B-D**) and following surgery (**C-E**).

6. I'm not sure that "less clonal T cells in tumor-adjacent normal lung" (line 266) can be taken to indicate a more tumor-focused T cell repertoire.

Response: We agree with *Reviewer 1* that this may have been an overstatement. We have revised this wording in the updated manuscript to state that (line 311):

"Overall, these findings suggest that the host's capacity to generate a stronger T cell response (as indicated by more T cells in PBMC) and a lower density and reactivity of T cells outside the tumor in the uninvolved tumor-adjacent lung (ie. bystander T cells) may be associated with better survival, while T cell responses targeting viral infections or shared mutations could hamper the immune system's ability to effectively combat the tumor."

7. An accounting of public TCRs (ie. which TCRs from which sample types are shared among 2 or more patients) would be a useful addition to the paper.

Response: We thank *Reviewer 1* for this constructive suggestion. As suggested, we have compiled a list of public TCRs. The findings are shown in **Reviewer Fig. RF6** (and **Fig. S15**) and show a large number of shared TCRs across patients both within and across tissue types. We have included a figure representing the number of shared TCRs across all tissues/patients (**Reviewer Fig. RF6A**), only blood samples (**Reviewer Fig. RF6B**), only uninvolved lung samples (**Reviewer Fig. RF6C**) or only tumors (**Reviewer Fig. RF6D**) in the manuscript. Furthermore, a list of TCRs found in ≥ 10 patients is now included as **Table S1A-D**. We used a cut-off of ≥ 10 patients because the number of TCRs shared by ≥ 2 patients was in the hundreds of thousands but are happy to provide the complete list if this is deemed to be of interest.

Reviewer Figure RF6. Substantial number of TCR sequences are shared across patients with early stage NSCLC. A) Shared CDR3 sequences across patients regardless of tissue type. **B)** Shared CDR3 sequences in blood of patients. **C)** Shared CDR3 sequences in the uninvolved adjacent lungs of patients. **D)** Shared CDR3 sequences in tumors from NSCLC patients. Grey bars represent the number of CDR3 shared in fewer than 10 NSCLC patients.

Reviewer #2

This manuscript describes T cell repertoire analysis for 236 early stage lung cancer tumors and adjacent lung. Their conclusion is that a "considerable proportion" of TCRs "appeared to be reactive to shared passenger mutations or viral infections". Higher sharing of TCRs in tumors and adjacent lung was associated with inferior survival. They indicate that this shows that "a concise understanding of shared antigens and T cells between tumor and adjacent normal tissue in NSCLC is needed to improve therapeutic efficacy and reduce risk of toxicity in the context of immunotherapy, particularly adoptive T cell therapy". The numerous enumerations and correlations of TCR features with each other and speculations as to their significance in this population are of modest interest, and there are other concerns.

8. The tumors were from resections from patients not treated with immunotherapy, which makes all conclusions relating to immunotherapy speculative.

Response: We agree with *Reviewer 2* regarding the speculative nature of comments pertaining to immunotherapy. As such, we have removed all mentions of immunotherapy from the *Results*, and only left them in our *Discussion* where we believe they are more appropriate as speculation highlighting potential implications of our findings.

9. The entire study is highly correlational and they seem to confuse correlation with causation throughout the manuscript and appear to use univariate analyses even when evaluating dozens of comparisons.

Response: We agree with *Reviewer 2* regarding the correlational nature of our study. As these studies were performed on archival samples, functional assays were not feasible. However, although the nature of this work is correlative, we do believe our findings offer novel insights to the field, as no such cohort of T cell receptor sequencing data has been made available to date on such a large cohort of patients with matched tissues, blood and supporting data from other analytical platforms. We also acknowledge that although univariate analyses were used for certain analyses, survival analyses underwent multivariate analyses to confirm the veracity of the associations described.

10. Not much attention is paid to potential regional heterogeneity in TIL within tumors.

Response: *Reviewer 2* is correct that no analyses accounted for regional heterogeneity of TILs in the current manuscript. Our prior work focused on characterizing this concept in a subset of 11 patients with early stage lung adenocarcinoma, and demonstrated that intratumor genomic and TCR heterogeneity play a key role in early stage lung cancer (Reuben *et al.* *Cancer Discovery*, 2017). In the present study, our goal was rather to evaluate the T cell repertoire both systemically and in the lungs of patients with NSCLC, to assess its association with other immunogenomic attributes and clinical outcome. We acknowledge that the regional heterogeneity of TILs could play a role. Unfortunately, as these samples were from a retrospective cohort, multiregion sequencing was not an option. However, to address *Reviewer 2*'s critique pertaining to the regional heterogeneity of TILs, we performed additional analyses based on whether T cells were enriched either at the tumor center or periphery to obtain some measure of T cell distribution. As shown in **Reviewer Fig. RF7** (and **Fig. S5**), we performed assessment of IHC samples to quantify CD3, CD4 and CD8 T cell density at the tumor center

and periphery. We then calculated a ratio of CD3, CD4, and CD8 T cells at the periphery and center to determine the T cell enrichment gradient and whether T cells may be in any way excluded from tumors (peripheral density > center density). As shown in **Reviewer Fig. RF7** (and **Fig. S5**), no differences are seen with CD3 or CD8 and T cell density (CD3, $p=0.1473$; CD8, $p=0.2394$), richness (CD3, $p=0.1324$; CD8, $p=0.4699$), and clonality (CD3, $p=0.1915$; CD8, $p=0.2132$). However, with CD4, though no difference was seen in T cell density ($p=0.0901$), T cell richness ($p=0.0006$) was significantly higher while clonality was significantly lower ($p=0.0120$) when CD4 were enriched at the tumor center. These findings could suggest different things. First, they suggest CD4 T cells may be the predominant population of T cells recruited to the tumor. This is supported by prior work by our group and others demonstrating the majority of lung tumors are enriched for CD4 rather than CD8 T cells (**Fig. S2B-C** and Reuben *et al.* *Cancer Discovery*, 2017). Secondly, CD4 T cells have the potential to inhibit T cell expansion and proliferation if they are regulatory (Treg). This means that an increased frequency of CD4 T cells at the tumor center could suppress CD8 T cell responses locally. Similar analysis performed on FoxP3 (data not shown), a marker associated with Tregs also transiently upregulated by activated CD8 T cells, demonstrated similar trends to those observed in CD4, though statistical significance was not attained, with higher richness ($p=0.0920$) and lower clonality ($p=0.1296$) in tumors with a higher FoxP3 density at the center. This supports the hypothesis that most CD4 in the tumor may have been Tregs and highlights the importance of assessing heterogeneity and spatial distribution of TILs. We have therefore included this figure in the updated manuscript (**Fig. S5**). In addition, we have added a sentence in the *Results* stating (line 135):

"Multi-region sequencing was not performed in this cohort, therefore we cannot exclude the possibility that intratumor heterogeneity may have played a role, as shown previously by our group and others."

Reviewer Figure RF7. T cell density, richness and clonality in tumors enriched for T cells at the tumor center or periphery. T cell density (A, D, G), richness (B, E, H) and clonality (C, F, I) in tumors with a higher density of CD3 (A-C), CD4 (D-F) or CD8 (G-I) in their center (white) or periphery (red).

11. No analysis of tumor differentiation with TIL/TCRs is done.

Response: As requested by Reviewer 2, we have performed additional analyses investigating the association between tumor differentiation patterns and the T cell repertoire. These analyses have shown that poorly differentiated tumors exhibit higher T cell clonality than their well ($p=0.0019$) or moderately ($p=0.0318$) differentiated counterparts (**Reviewer Fig. RF8 and Fig. S10**). These data suggest that poorly differentiated tumors may be more immunogenic than their counterparts, thereby allowing T cell antigen recognition and expansion. The figure has been added to the *Results* and reads as such (line 192):

“T cell density and richness showed no differences based on tumor differentiation, though poorly differentiated tumors did exhibit higher T cell clonality than well and moderately differentiated tumor ($p=0.0019$ and $p=0.0318$, respectively; Fig. S10A-C).”

Reviewer Figure RF8. T cell clonality is increased in poorly differentiated tumors. Comparison of **A)** T cell density, **B)** richness, and **C)** clonality in well, moderately, and poorly differentiated tumors.

12. Many of the correlations (such as 4C, and S10F, G, H) are marginally significant and completely unconvincing visually.

Response: As pointed out by *Reviewer 2*, these correlations were modest though statistically significant. In the case of the correlation between shared mutations and shared TCRs between tumor and uninvolved lungs, the modest correlation could suggest other factors contribute to this phenomenon, such as the specificity of shared T cells for viral antigens presented in the same figure. In order to avoid overstating these findings, we have reworded our *Results* to "modest" (line 261) in the text. We have also revised our language in the *Results* for former **Fig. S10F-H** (now **Fig. S16F-H**) which now reads as such (line 260):

*"In regards to the T cell repertoire, a higher proportion of mutations unique to the tumor was modestly associated with a higher T cell clonality in the tumor ($r=0.22$; $p=0.028$, **Fig. S16F**), while more unique mutations in the uninvolved tumor-adjacent lung ($r=-0.23$; $p=0.027$) or more shared mutations ($r=-0.20$; $p=0.048$) was associated with lower tumor T cell clonality (**Fig. S16G-H**)."*

13. The GLIPH program described in reference 32 was really only carefully tested against tuberculosis antigens, and the "viral T cell responses" argument is also speculative.

Response: Reviewer 2 is correct that the virally related T cell repertoires were indeed speculative in nature. The GLIPH algorithm was designed to robustly infer T cell specificities by grouping CDR3 beta chain repertoires together if sharing similar motifs. In the case of flu-related motifs, for example, very detailed mapping of residues on the TCR side that are needed to engage with antigenic peptides was made possible with crystallography data as well as CDR3 sequences of tetramer-specific sorted T cells. From the crystallography studies on the flu M1 peptide, we have learned that there are at least two most prevalent motifs, namely "RS" and "GxY", within the beta chain CDR3 region [redacted]. Consistently, using A02-M1 tetramer-sorted CD8 T cell CDR3 sequences, the GLIPH algorithm has found predominantly motifs with these two sequences that are defined by the crystal data. Therefore, though to date GLIPH has only truly been validated on tuberculosis antigens, we have been able to accurately infer virally-related specificities with tetramer-derived CDR3 sequences by using GLIPH and therefore feel it is an important computational tool. However, in order to avoid overstating our conclusions, we have reworded the *Results* (line 271):

“Accordingly, we studied TCR motifs and their antigenic specificity using the GLIPH algorithm, a computational tool validated on tuberculosis antigens utilized to predict antigen binding based on comparison of TCR sequencing data to tetramer-validated sequences to identify shared amino acid motifs and infer antigen specificity.”

[redacted]

14. All of the survival correlations in figure 5 are of marginal statistical significance, and T-cell density has long been shown to be correlated with survival.

Response: We agree with *Reviewer 2* that p-values presented in **Fig. S18A-C** (formerly **Fig. 4A-C**) offer modest statistical significance. However, we believe a difference in survival in a localized lung cancer cohort with comprehensive clinical annotation based solely on TCR parameters remains of value. Furthermore, though T cell density in the *tumor* has been shown to be associated with survival and response, **Fig. S18A** (formerly **Fig. 4A**) depicts T cell density in the *peripheral blood*, based on TCR sequencing, and its association to outcome. To our knowledge, this has not been shown to date in non-small cell lung cancers and if it has, we would be more than happy to cite this reference as corroboration. These analyses also underwent multivariate testing to adjust for any confounding parameters and confirm statistical significance. Therefore, we believe these data are of interest, particularly considering the greater accessibility and limited invasiveness of blood-based assays. However, in order to de-emphasize these

findings in light of the modest statistical significance, we have moved them to **Fig. S18A-C** along with the lung cancer-specific survival plots requested in query #15.

15. Also, as the median time to recurrence after surgery is 11 months and the median survival after metastatic relapse is 6 to 8 months, the figures showing overall survival out to 15 years suggest that most of the deaths in this group were not from lung cancer relapse, but rather other causes. Lung cancer specific mortality should be shown.

Response: Reviewer 2 raises an important point as our prolonged follow up could have confounded survival analyses due to deaths unrelated to lung cancer. As requested, we have reviewed and obtained lung cancer-related death information from patient medical charts, and repeated these analyses focusing exclusively on confirmed lung cancer-related deaths – cases for which there was any doubt as to cause of death were excluded. These analyses are shown in **Reviewer Fig. RF10** (and **Fig. S18D-F**) and show similar trends to those observed with overall survival, though the decrease in sample size may hamper statistical significance. We have also added language to the *Results* (line 301):

*"Analysis of lung cancer-specific survival revealed much the same trends though smaller numbers may have limited statistical significance ($p=0.0717$, $p=0.1428$, and $p=0.0511$, respectively; **Fig. S18D-F**)."*

Reviewer Figure RF10. Lung cancer-specific survival based on T cell repertoire analysis in the peripheral blood and uninvolved adjacent lung. Overall survival based on **A)** T cell density in the peripheral blood, **B)** T cell density in the adjacent uninvolved lung, and **C)** T cell clonality in the adjacent uninvolved lung-enriched (vs tumors) T cell repertoire. Red, above median; Blue, below median. Lung cancer-specific survival based on **D)** T cell density in the peripheral blood, **E)** T cell density in the adjacent uninvolved lung, and **F)** T cell clonality in the adjacent uninvolved lung-enriched (vs tumors) T cell repertoire. Red, above median; Blue, below median.

16. Adjacent lung is not "normal", and this word should be removed from the description.

Response: Reviewer 2's point is well taken. As suggested, we have replaced all mentions of "normal lungs" with "tumor-adjacent uninvolved lungs".

17. "NSEM" is not defined in its first use in the figure legend to S4.

Response: We apologize for this omission, we have now defined NSEM (non-synonymous exonic mutations) in the legend to former **Fig. S4** (now **Fig. S6**).

18. In S5H it would be more interesting to look at clonality in EGFR mutants with a high TMB rather than low.

Response: We thank *Reviewer 2* for this constructive suggestion. We understand *Reviewer 2's* concern in looking at $EGFR_{MUT}$ patients with a low TMB, considering a higher TMB is generally associated with greater immunogenicity and better outcome. However, in our cohort, TMB in $EGFR_{MUT}$ patients was always below 76 NSEM and thus confined to the lower tertile of all studied patients (**Fig. S8A**). As such, it made it impossible to evaluate $EGFR_{MUT}$ tumors harboring a high TMB. This is in line with other studies (Offin *et al.*, CCR, 2019) which showed a lower TMB in $EGFR_{MUT}$ tumors. However, as requested by *Reviewer 2* and in order to assess the relationship between TMB and T cell repertoire within $EGFR_{MUT}$ patients, we have now performed correlations between TMB and T cell clonality only in $EGFR_{MUT}$ tumors as shown in **Reviewer Fig. RF11** (and **Fig. S9**). We have also divided $EGFR_{MUT}$ tumors into high and low TMB (median) and compared T cell density, richness and clonality. These analyses reveal no clear trends or differences between TMB and T cell repertoire, though this could be limited by our number of $EGFR_{MUT}$ tumors. We now include these analyses as **Fig. S9A-F** in the revised manuscript, along with text in the *Results* section (line 183):

"Of note, even within the highest TMB EGFR_{MUT} tumors, no differences were observed in T cell repertoire attributes (Fig. S9A-F). Taken together, these results suggest that there exist TMB-independent mechanisms contributing to the low clonality in EGFR_{MUT} NSCLC tumors."

Reviewer Figure RF11. T cell repertoire features are not associated with higher tumor mutational burden among EGFR-mutant patients. Comparison of TMB^{hi} (above median, white) or TMB^{lo} (below median, red) tumors relative to **A**) T cell density, **B**) T cell richness, and **C**) T cell clonality. Correlation between **(D-F)** tumor mutational burden and **D**) T cell density, **E**) richness, and **F**) clonality only in tumors harboring classical EGFR mutations.

Reviewer #3

The authors thoroughly evaluated T cell repertoire based on TCR sequencing of CDR3 variable regions in 236 resected non-small cell lung cancer (NSCLC) tumors, and their matched tumor-adjacent normal lung as well as peripheral blood. These tumor genomic and microenvironment profiles were reported in Karada et al. (2017). They assessed the associations between T cell repertoire features (diversity and clonality) and tumor genomic, microenvironment, and clinicopathological aspects. They identified that characteristics of T cell repertoire in tumor-adjacent normal tissue were distinguished from non-cancer normal lung and highly associated to nearby tumors. Furthermore, the authors found the patient outcome was associated with T cell clonality in tumor-adjacent normal tissue and density of T cells in PBMC.

This is a well-designed study providing insight into the process of immune editing in early stage of NSCLC, as well as the impact in relation to neighboring tissues. The abnormality of T cell repertoires in the tumor-adjacent normal tissue suggests that future consideration should be focused on understanding and harnessing the activity of increased anti-tumor T cells that have repertoires that are distinct from adjacent normal regions. However, there are several minor points that authors should address to maximize the impact of the study.

19. The authors mentioned that the genomic profiles of the cohort were reported in Karada et. al (reference 23). However, Karada et al. reported whole exome sequencing data of 108 tumor and normal pairs. The authors should clearly indicate how many tumor samples in this study have genomic data profiles, and were subjected to the analyses exploring the relationship between T cell repertoire and genomic features (TMB, driver genes).

Response: We apologize for the lack of clarity in our description of samples and analyses and our omission of a critical reference. In addition to reference 23 by Kadara, the samples included in this study contained those published by Choi (Choi *et al.*, Annals of Oncology, 2017). We have added this reference and included a summary table which lists the number of samples (blood, uninvolved adjacent lung, tumor) having undergone each assay/analysis. The table is shown below for convenience (**Reviewer Table RT2** and **Table S2**) and included in the updated manuscript.

Reviewer Table RT2. Overview of samples and assays performed.

Assay	Tissue type		
	Blood	Uninvolved Lung	Tumor
Whole exome sequencing	96	215	215
RNA microarray	---	---	141
Immunohistochemistry	---	---	146
TCR sequencing	120	216	236
GLIPH	---	168	168

20. The authors observed that T cell clonality was highly correlated to cytotoxic phenotype (i.e. CD8 T cells, GZMB positive cells) and TMB. However, they also found that TMB was highly associated with GZMB positive cells. Therefore, it is essential to perform

multivariate analysis to determine if the number of GZMB positive cells and TMB (in log scale) are two independent factors that determine T cell clonality.

Response: We thank *Reviewer 3* for this suggestion. As requested, we have performed multivariate analysis to determine the contribution of GZMB and TMB to T cell clonality. We first tested whether TMB alone and TMB + GZMB had the same effect on predicting T cell clonality using an ANOVA function. These analyses revealed that adding GZMB led to a significant improvement over using TMB alone for predicting T cell clonality ($p=0.0094$). We then performed the same analysis focusing on GZMB with or without TMB to predict T cell clonality using the same method. These analyses demonstrated that adding TMB led to a marginal though significant improvement over simply using GZMB to predict T cell clonality ($p=0.04$). Therefore, we conclude that TMB and GZMB both contribute independently to predicting T cell clonality.

21. The Kruskal-Wallis test has been utilized to assess the differences among groups (i.e. Figure 3). However, the TCR variables (e.g. clonality) are paired (i.e. tumor and matched adjacent normal), and vary over a wide range, it would be more appropriate to utilize a paired test (i.e. Wilcoxon signed-rank test) to confirm the differences between compartment pairs.

Response: We agree with *Reviewer 3's* assessment that a paired test is better suited for comparisons between tissue compartments in **Fig. 3**. As such, we have repeated statistical analyses using a Wilcoxon matched-pairs signed rank test. The updated p -values are shown in **Reviewer Fig. RF12** (and **Fig. 3**).

Reviewer Figure RF12. T cell clonality is highest in the uninvolved lungs of NSCLC patients. A) T cell density, B) T cell richness, and C) T cell clonality in peripheral blood, tumor-adjacent uninvolved lungs and tumors from patients with NSCLC.

22. A more comprehensive approach would be achieved if the analyses of T cell repertoire in the adjacent normal (Figure S8C-E) were based on comparisons of four groups (healthy, COPD, never-smokers and smokers) to distinguish the effects of cancer and smoking to adjacent normal regions.

Response: Reviewer 3 makes an excellent point. As requested, we have revised this analysis to distinguish smoker from non-smoker tumor-adjacent uninvolved lungs from patients with NSCLC. The analysis is presented in **Reviewer Fig. RF13** (and **Fig. S13**) and reveals that uninvolved tumor-adjacent lungs from NSCLC patients largely behave similarly whether patients were smokers or non-smokers, but distinctly from healthy lungs. This new analysis is now included in the manuscript as **Fig. S13**. We've also added text to the *Results* (line 230):

"Interestingly, richness was lowest ($p < 0.0001$) while clonality was highest ($p < 0.0001$) in the uninvolved tumor-adjacent lungs of smokers and non-smokers, highlighting a more active antigenic response that could be related to the tumor (Fig. S13B-C)."

Reviewer Figure RF13. Tumor-adjacent uninvolved lungs exhibit consistent T cell repertoire attributes irrespective of smoking status. A) T cell density, B) T cell richness, and C) T cell clonality in healthy lungs, lungs from COPD patients, and tumor-adjacent uninvolved lungs from smokers and non-smokers with NSCLC.

23. On page 10 and Figure S9B-C, “lung-enriched T cell repertoire” was mentioned. The authors should clarify how this is defined. Are they top 100 T cell clones detected in the specific compartment?

Response: We apologize for the lack of clarity in our description of the analysis of lung-enriched T cells. This analysis was based on comparison of the entire repertoire in both compartments to identify clones which are statistically-enriched in one versus the other. Subsequent analyses then focused exclusively on these statistically-enriched T cells. The parameters for this analysis are now outlined in the *Methods* section and read as such (line 402):

"To identify TCRs that were enriched in one tissue compared to another, we applied a differential abundance framework as described previously (REF). Parameters were as follows: minTotal=5, productiveOnly=True, alpha=0.1, count=aminoAcid."

24. The tumor-adjacent normal regions have low T cell infiltration and diversity (Figure 4A-B), but higher clonality (Figure 4C) compared to matched tumors. On page 9 the authors suggest that there are high antigenic responses in tumor-adjacent normal

regions. However, T cell clones between tumor and adjacent normal were highly overlapped (Figure 4B). Together these results also imply that specific T cell clones detected in the adjacent normal regions are prevalent tumor-target T cell clones targeting shared mutations in both tumor and adjacent normal tissue. These T cells in the adjacent normal tissues might reflect immune exhaustion and suppression in nearby tumor areas. This could be another explanation for the association between the poor survival and high level of T cell density and clonality in adjacent normal tissues (Figure 5). The findings are very much in line with, and support, the rich literature documenting an immune suppressed tumor microenvironment in NSCLC.

Response: We appreciate Reviewer 3's explanation regarding our findings and agree that this may be a possible explanation. As such, we have now included this explanation in the *Results* which reads as such (line 223):

"These findings could reflect bystander T cell reactivity in the adjacent uninvolved lungs as recently described, or an accumulation of exhausted tumor-reactive T cells outside the tumor microenvironment."

REVIEWERS' COMMENTS:

Reviewer #1 (Remarks to the Author):

My initial comments have been adequately addressed and I believe the additional analysis has strengthened the paper. In my opinion the revised manuscript is suitable for publication, but I would ask that the authors reconsider how they present the results of one of their new results. They indicate that "Similar analysis performed on FoxP3 ...demonstrated similar trends to those observed in CD4, though statistical significance was not obtained". Without statistical significance the authors shouldn't assert that this result "supports the hypothesis that most CD4 in the tumor may have been Treg".

Reviewer #2 (Remarks to the Author):

The comments raised in the prior review were all comprehensively addressed and appropriate changes made.

David Carbone

Reviewer #3 (Remarks to the Author):

The authors have addressed the reviewers' questions and critiques.

Tuesday, October 29th 2019

Tanya Bondar, Ph.D.
Senior Editor
Nature Communications

re: Resubmission of manuscript to *Nature Communications*

Dear Dr. Bondar,

We respectfully resubmit our manuscript entitled "*Comprehensive T cell repertoire characterization of localized non-small cell lung cancer*" to *Nature Communications*. You'll find below a point-by-point response to reviewer comments.

Reviewer #1

My initial comments have been adequately addressed and I believe the additional analysis has strengthened the paper. In my opinion the revised manuscript is suitable for publication, but I would ask that the authors reconsider how they present the results of one of their new results. They indicate that "Similar analysis performed on FoxP3 ...demonstrated similar trends to those observed in CD4, though statistical significance was not obtained". Without statistical significance the authors shouldn't assert that this result "supports the hypothesis that most CD4 in the tumor may have been Treg".

Response: We understand the reviewer's point. As this was only included in the reviewer response, it is not an issue to be addressed in the current version of the manuscript.

Reviewer #2

The comments raised in the prior review were all comprehensively addressed and appropriate changes made.

Response: Thank you for your thoughtful comments and critiques.

Reviewer #3

The authors have addressed the reviewers' questions and critiques.

Response: Thank you for your thoughtful comments and critiques.

Once again, we appreciate the thoughtful review of our manuscript for publication in *Nature Communications*.

Please do not hesitate to contact me should you require any additional information.

Sincerely,

Jay Zhang, M.D., Ph.D.